# The Kosterlitz-Thouless phase transition: an introduction for the intrepid student

Victor Drouin-Touchette*

Center for Materials Theory, Rutgers University, Piscataway, New Jersey 08854, USA
* vdrouin@physics.rutgers.edu

July 29, 2022

## Abstract

**This is a set of notes recalling some of the most important results on the XY model from the ground up. They are meant for a junior research wanting to get accustomed to the Kosterlitz-Thouless phase transition in the context of the 2D classical XY model. The connection to the 2D Coulomb gas is presented in detail, as well as the renormalization group flow obtained from this dual representation. A numerical Monte-Carlo approach to the classical XY model is presented. Finally, two physical setting that realize the celebrated Kosterlitz-Thouless phase transition are presented: superfluids and liquid crystal thin films.**

## 1   Introduction

In this set of notes we will review some essential concepts pertaining to the Kosterlitz-Thouless phase transition. Although I have attempted to be as exhaustive as possible, this subject is incredibly vast. We review the essential theory and numerical aspects of the classical XY model in a manner accessible for the advanced undergraduate and graduate students.

Starting from a two-dimensional model of planar spins interacting ferromagnetically (XY model), we will show that even though the model cannot exhibit long-range order (LRO), it can have quasi-long-range order (QLRO) with algebraically decaying correlations. We will formalize how one can transition between disorder and QLRO through the introduction of vortices. We will derive the renormalization group equations (RG) from the dual Coulomb gas model. We will then delve into the numerical Monte-Carlo approach to simulate the XY model. We will finish by illustrating different physical situations where KT physics is at play due to their dimensionality and planar symmetry. As it applies to a large set of disjointed physical contexts, the Kosterlitz-Thouless phase transition is perhaps one of the best example of universality.

The XY model is described through a classical hamiltonian

$$H = -J \sum_{\langle i,j \rangle} \boldsymbol{S}_i \cdot \boldsymbol{S}_j = -J \sum_{\langle i,j \rangle} \cos\left[\theta_i - \theta_j\right], \tag{1}$$

where the sum is over all pairs of nearest neighbors, *i.e.* bonds on the square lattice. In the second form of the hamiltonian, we took advantage of the $O(2)$ symmetry (hence the XY name - spins lie in the XY plane), such that $\boldsymbol{S}_i = (\cos\theta_i, \sin\theta_i)$. Owing to the ferromagnetic interaction $J$, at $T = 0$, the system would be in an ordered state where $\theta_i = \theta_0$ for all $i$. What is the fate of such a state in the presence of thermal fluctuations?

The Mermin-Wagner theorem [1] states that "*in $d \leq 2$, no stable ordered phase at finite temperature can exist if the system is invariant under a continuous symmetry*". Well, there goes our luck. There can be no long-range order in the 2D XY model! This is probably disappointing, as in the Ising model in 2D [2], where spins can only take the discrete values $\sigma_i = \pm 1$, an exact solution by Onsager shows that there is a critical temperature $T_c = 2J/\log(1 + \sqrt{2})$. Below this temperature there is long-range order.

The feat of Berezinskii, Kosterlitz and Thouless [3–5] in the late 1970s was to show that this model instead shows quasi-long-range order, and to point out directly the mechanism by which

this ordered phase sets in. This transition is peculiar - it does not fall in any second or first order universality class. Instead, it is referred as an infinite-order phase transition.

These notes take inspiration from a few other written works; some famous books [6,7], other less famous [8] but as full of wisdom, and some review articles [9]. The avid reader is encouraged to go through these.

## 2 Algebraic order in the 2D XY model

In a low-temperature expansion, the angle difference between two spins will be small: $|\theta_i - \theta_j| \ll 2\pi$. In this small fluctuation regime, we can approximate the cosine term in the hamiltonian to extract the long-range behavior.

$$
\begin{aligned}
H &= -J \sum_{\langle i,j \rangle} \cos(\theta_i - \theta_j) \\
&= -JN + \frac{J}{2} \sum_{\langle i,j \rangle} (\theta_i - \theta_j)^2 \\
&= E_0 + \frac{J}{4} \sum_{r,a} (\theta(r + a) - \theta(r))^2 \\
&\simeq E_0 + \frac{J}{2} \int d^2 r (\nabla \theta(r)^2 .
\end{aligned}
\tag{2}
$$

In the last line, we have taken the continuum limit, and replaced the field $\theta_i$ by a continuous one, $\theta(r)$, as slowly varying function of $r$. From this, we can extract a lot of information about the magnetization and correlation functions.

### 2.1 Average magnetization

We calculate the average magnetization in the $x$ direction for the 2D XY model ($y$ is identical). We have:

$$
\langle S_x \rangle = \langle \cos \theta(r) \rangle = \langle \cos \theta(0) \rangle
\tag{3}
$$

$$
= \frac{\text{tr}_{\{\theta_i\}} \cos \theta(0) e^{-\beta H}}{\text{tr}_{\{\theta_i\}} e^{-\beta H}}
\tag{4}
$$

$$
= \text{Re}\left( \frac{1}{\mathcal{Z}} \int \mathcal{D}[\theta_i] \cos(0) e^{-\beta H + \iota \theta(0)} \right),
\tag{5}
$$

where $\mathcal{Z}$ is the partition function $\text{tr}_{\{\theta_i\}} e^{-\beta H}$, and in the first line, we took advantage of translation invariance to set the spin at site $r = 0$. In order to calculate that expression, we Fourier transform the $\theta$ variable, with periodic boundary conditions. We then have

$$
\theta(r) = \frac{1}{\sqrt{N}} \sum_k \theta_k e^{\iota k \cdot r} ,
\tag{6}
$$

and therefore, the hamiltonian in momentum space (with $a$ being the lattice spacing) becomes

$$H = E_0 + \frac{JS^2 a^2}{2} \sum_k k^2 \theta_k \theta_{-k}$$

$$= E_0 + JS^2 a^2 \sum_k{}' k^2 (\alpha_k^2 + \beta_k^2). \tag{7}$$

In the second line, we decomposed the Fourier modes $\theta_k$ into $\theta_k = \alpha_k + \iota \beta_k = (\theta_{-k})^*$, and the primed sum now runs over half of the Brillouin zone. This leads to, after some algebra:

$$\langle S_x \rangle = \exp\left(-\frac{T}{2J} I(L)\right), \tag{8}$$

with $I(L)$ a geometric factor written as

$$I(L) = \int_{\pi/L}^{\pi/a} \frac{dk}{k} = \ln\left(\frac{L}{a}\right). \tag{9}$$

We see that $I(L)$ will blow up to infinity as $L$ reaches the thermodynamic limit $L \to \infty$, and then, for any $T \neq 0$, the logarithmic divergence of this geometric factor will force $\langle S_x \rangle$ to 0. This is directly the statement of the Mermin-Wagner theorem. Hence there can be no ordered low-temperature phase (in the conventional long-range order) in the 2D XY model.

## 2.2 Correlation functions

We now set on the same path, but for the spin-spin correlation function:

$$g(r) = \langle \exp\{\iota(\theta(r) - \theta(0))\}\rangle = \frac{\mathrm{tr}_{\{\theta_i\}} e^{\iota(\theta(r) - \theta(0))} e^{-\beta H}}{\mathrm{tr}_{\{\theta_i\}} e^{-\beta H}}. \tag{10}$$

Again, using the Fourier transform of the Hamiltonian and the decomposition of the Fourier modes, we get

$$\mathrm{tr}_{\{\theta_i\}} e^{\iota(\theta(r) - \theta(0))} e^{-\beta H}$$

$$= \prod_k \int \mathcal{D}[\alpha_k] \int \mathcal{D}[\beta_k] e^{\iota(\theta(r) - \theta(0))} e^{-\beta E_0 - \beta J a^2 \sum_k{}' k^2 (\alpha_k^2 + \beta_k^2)}$$

$$= \prod_k \int \mathcal{D}[\alpha_k] \int \mathcal{D}[\beta_k] \exp\left(\frac{Ja^2}{k_B T} \sum_k{}' k^2 (\alpha_k^2 + \beta_k^2) + \frac{\iota}{\sqrt{N}} \sum_k \theta_k (e^{\iota k \cdot r} - 1)\right) \tag{11}$$

$$= \prod_k \int \mathcal{D}[\alpha_k] \exp\left(\frac{Ja^2}{k_B T} \sum_k{}' k^2 \alpha_k^2 + \frac{\iota}{\sqrt{N}} \sum_k \alpha_k (e^{\iota k \cdot r} - 1)\right)$$

$$\times \int \mathcal{D}[\beta_k] \exp\left(\frac{Ja^2}{k_B T} \sum_k{}' k^2 \beta_k^2 - \frac{1}{\sqrt{N}} \sum_k \beta_k (e^{\iota k \cdot r} - 1)\right),$$

where the measured spins $\theta(r) - \theta(0)$ have also been Fourier transformed. Performing the two Gaussian integrations over the fields $\alpha_k$ and $\beta_k$ leads to

$$\mathrm{tr}_{\{\theta_i\}} e^{\iota(\theta(r) - \theta(0))} e^{-\beta H} = \prod_k \sqrt{\frac{\pi k_B T}{2J a^2 k^2}} \exp\left(-\frac{k_B T (e^{\iota k \cdot r} - 1)(e^{-\iota k \cdot r} - 1)}{2J a^2 k^2 N}\right). \tag{12}$$

A similar calculation for the denominator gives the same expression, but with $r = 0$ set. The final result is therefore

$$g(r) = \exp\left(-\frac{k_B T}{NJa^2} \sum_{\boldsymbol{k}} \frac{1 - \cos(\boldsymbol{k} \cdot \boldsymbol{r})}{k^2}\right). \tag{13}$$

When interested in the long-range behavior, one may change the discrete momentum sum for a continuous integral $1/N \sum_{\boldsymbol{k}} \rightarrow (a/2\pi)^2 \int d^2 k$. The integral can then be performed after a polar coordinate change, and we get

$$g(r) = \exp\left(-\frac{k_B T}{2\pi J} \int_0^{\pi/a} dk \frac{1 - J_0(kr)}{k}\right) \tag{14}$$

$$= \exp\left(-\frac{k_B T}{2\pi J} I(r)\right) \tag{15}$$

$$\text{with} \tag{16}$$

$$I(r) = \int_0^{\pi r/a} dx \frac{1 - J_0(x)}{x}, \tag{17}$$

with $J_0$ the zeroth-order Bessel function. At long distances $r \gg a$, the Bessel contribution to $I(r)$ is negligeable and we can approximate $I(r) \sim \ln(\pi r/a)$, which leads to

$$g(r) \approx \exp\left(-\frac{k_B T}{2\pi J} \ln \frac{\pi r}{a}\right) = \left(\frac{\pi r}{a}\right)^{-k_B T/2\pi J} = \left(\frac{\pi r}{a}\right)^{-\eta(T)}. \tag{18}$$

We see from this equation that the correlation function falls off algebraically at all finite temperatures. Interestingly, we can compare this with the expected behavior of correlation functions in disordered phases

$$g(r) \approx \frac{\exp(-r/\xi(T))}{r^{d-2+\eta}}, \tag{19}$$

and we can conclude from this that, at low-temperatures, the XY model has an infinite correlation length $\xi = \infty$, and $\eta$ is temperature dependent. This means that, at all low-temperatures, the system is critical. Furthermore, one can extract the correlation function at high temperature in a high-temperature expansion of the partition function of the XY model. This leads to

$$g(r) == \left(\frac{\beta J}{2}\right)^{r/a} = e^{-r/\xi}, \tag{20}$$

where the correlation length is $\xi = a/\ln(2T/J)$. This exponential decay is the signal that all ferromagnetic order is destroyed in the system, and the system is disordered. Clearly, something is ongoing in this system that leads to some phase transition. The insight of Berezinskii [3] in 1971 and independently of Kosterlitz and Thouless [4,5] in 1973 was to consider another type of excitation in the system, different from the typical spin wave excitations. Looking past the usual spin-waves, the authors found that the $O(2)$ symmetry was consistent with vortex-like excitations, defects in the phase around which the field $\theta$ winds by $2\pi$.

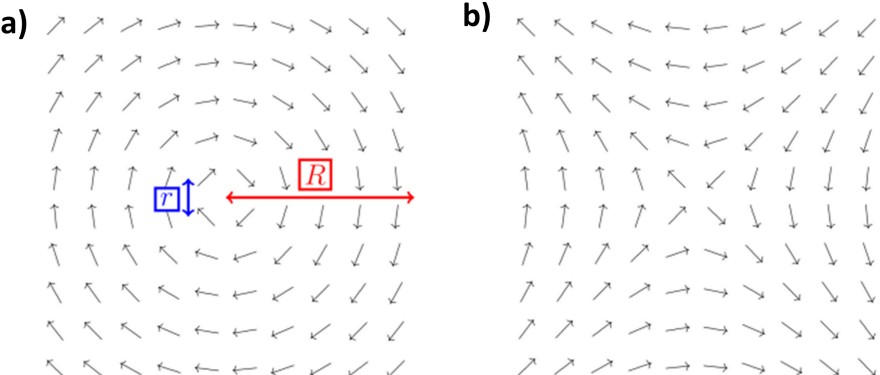

Figure 1: (a) Following the vectors around the plaquette in a counterclockwise way, the vectors turn $2\pi$ while we circle $2\pi$. This is a vortex. Its core radius is $r$, therefore the energy of such a vortex is $E \sim \ln(R/r)$. In (b), the vectors wind by $-2\pi$ while we circle counterclockwise - this is an antivortex. Adapted from Ref. [10].

## 2.3 Vortices and entropic argument

Vortices are topological defects of the field $\theta(\boldsymbol{r})$, satisfying the Laplace equation $\nabla^2\theta(\boldsymbol{r}) = 0$. Apart from the trivial solution to this equation ($\theta(\boldsymbol{r}) = 0$, the ferromagnetic ground state), there are solutions called vortices. For a single vortex situated at $\boldsymbol{r}_0$, the circulation loop integral around it needs to be quantized:

$$\oint_{\boldsymbol{r}_0} \nabla\theta(\boldsymbol{r}) \cdot d\boldsymbol{l} = 2\pi n \,, \tag{21}$$

with $n < 0$ corresponding to clockwise winding vortices, and $n > 0$ to anticlockwise. Such configurations are illustrated in Fig. 1.

Can the proliferation of these objects be the culprit for the loss of QLRO? To estimate this, we consider the cost to the free energy $\Delta F = \Delta E - T\Delta S$ of adding a free vortex a system with no vortex in it. In order to estimate the energy generated by the presence of an isolated vortex, we must first estimate $\nabla\theta$. We use our equation 21, from which we estimate that, if there is **one** vortex on the lattice, then $\nabla\theta = \frac{n}{r}\widehat{\theta}$. Therefore, the energy difference associated with this isolated vortex is

$$\Delta E = \frac{J}{2}\int d^2r(\nabla\theta(\boldsymbol{r}))^2 = \pi J n^2 \int_a^L \frac{dr}{r} = \pi J n^2 \ln\frac{L}{a} \,, \tag{22}$$

with $L$ being the linear dimension of the system. We note that is a truly continuous system, we would have to start the integral at 0. However, our integral would then be divergent. It is therefore important here to consider the fact that all of this truly takes place on a lattice, where we have a lower spatial bound to this integral, *i.e.* the lattice constant $a$.

We then calculate the entropic cost to the creation of a vortex. We have that $\Delta S = k_B \ln\Omega$, with $\Omega$ being the number of microstates that can be occupied with one vortex. Since we work on a lattice of size $L^2$ with a lattice constant $a$, this means there are $(L/a)^2$ ways to put this one vortex

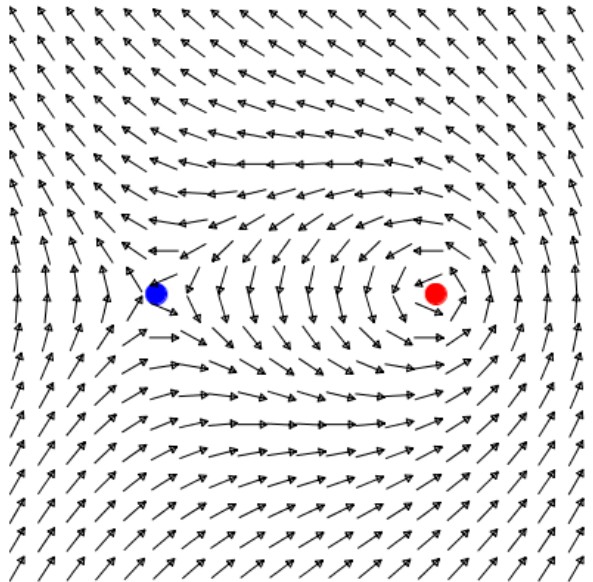

Figure 2: Vortex pair configuration.

on the lattice. Hence, we have:

$$\Delta S = k_B \ln (L/a)^2 = 2k_B \ln L/a \ . \tag{23}$$

Hence, the cost in free energy to the creation of an isolated vortex is, in this heuristic approximation,

$$\Delta F = \Delta E - T\Delta S = (\pi J n^2 - 2k_B T) \ln \frac{L}{a} \ . \tag{24}$$

We can clearly see the following two regimes:

- For $k_B T < \pi J/2$, $\Delta G > 0$, and then isolated vortices are unfavourable. If they exist at all in the system, it will be in neutral pairs, where their effect at long distance is negated. Such a bound configuration is shown in Fig. 2.

- For $k_B T > \pi J/2$, $\Delta G < 0$, and then isolated vortices are favourable and proliferate.

This provides us with our first crude estimate for the KT transition: $k_B T_c = \pi J/2$. We can now say that it is the unchecked proliferation of free vortices that kills the quasi-long-range order and leads to disorder. This remarkably simple argument from Kosterlitz and Thouless is not too far from the truth; one has to include the effect of the screening of ambient vortex pairs in the system to the interaction strength $J$ to get a faithful and complete picture. To further probe this mechanism, we need to map the spin model to that of the 2D Coulomb gas and proceed with a renormalization group analysis.

## 3   Renormalization-group analysis

To proceed further in our analysis, we need to map the 2D classical XY model's partition function to the partition function of a 2D Coulomb gas. When the exchange of photons is strictly confined

to two dimensions, the Coulomb interaction between two charges does not behave as $q_1 q_2 / r$ with $r$ being the distance between two charges, but as $q_1 q_2 s \log(r)$. We will see how this model emerges naturally from the XY model when vortices are taken into consideration. From the 2D Coulomb gas, we will then derive an effective Hamiltonian for two charges in the presence of two other charges, thereby describing the effect of screening of these charges. This will lead us to a renormalization group description of the XY model.

## 3.1   Mapping to the Coulomb gas

Starting from the reduced Hamiltonian, $\mathcal{H} = -\beta H$, that we can write in the continuum limit as (neglecting the constant ferromagnetic ground state energy):

$$\mathcal{H} = -\frac{K}{2} \int d^2 r |\nabla \theta(\boldsymbol{r})|^2 \,, \tag{25}$$

with $K = \beta J = J/k_B T$. We also remember the important condition on the $\theta$ field which allows the creation of vortices, namely the quantization of the loop integral around a vertex (see eq. 21). We note, that for a vortex of charge $n = 1$, the simplest ansatz for the $\theta$ field created is $\theta(\boldsymbol{r}) = \arctan(y/x)$, which leads to $\nabla \theta = (-y/r^2, x/r^2)$. This gives the same expression for the estimate of the energy of a solitary vortex as in eq. 22, $\Delta E = J\pi \ln(L/a)$. Our goal is to make this equation more tractable, and to isolate the two contributions to the fluctuations in the system: a Gaussian one, coming from the spin-waves in the ordered states, and a topological one, concerning the vortices.

First of all, we see that our long range action concerns a kind of velocity field for a fluid, $\boldsymbol{u} = \nabla\theta$. What we wish to do is decompose this field into a sum of two terms. The first is a flow in which there are no vortices, $\boldsymbol{u}_0 = \nabla\phi$ with $\phi$ a scalar function such that $\nabla \times \boldsymbol{u}_0 = 0$, *i.e.* there is no vortices. The second term would then include the vorticity charge. We first rewrite the circulation integral around a $n$-charge vortex.

$$2\pi n = \oint \boldsymbol{u} \cdot d\boldsymbol{l} = \int d^2 x \widehat{z} \cdot (\nabla \times \boldsymbol{u}) \,. \tag{26}$$

We then see that setting $\nabla \times \boldsymbol{u} = 2\pi\widehat{z}\sum_j n_j \delta(\boldsymbol{r} - \boldsymbol{r}_j)$ solves this equation. This term sets the position $\boldsymbol{r}_j$ of vortices of charge $n_j$, hence the integral over a certain area counts all the vortex charges present inside, which gives rise to the $2\pi n$ term. Having set that, we write our decomposition of the velocity field, using another scalar field $\psi$:

$$\boldsymbol{u} = \boldsymbol{u}_0 - \nabla \times (\widehat{z}\psi) = \nabla\phi - \nabla \times (\widehat{z}\psi) \,. \tag{27}$$

We see that $\nabla \times \boldsymbol{u} = -\nabla \times \nabla \times (\widehat{z}\psi) = \widehat{z}\nabla^2\psi$. Using our previous setting of the counting of vortices, we recover a Poisson equation in 2D with a potential due to point charges $2\pi n_j$:

$$\nabla^2\psi = 2\pi \sum_j n_j \delta(\boldsymbol{r} - \boldsymbol{r}_j) \,, \tag{28}$$

and its solution:

$$\psi(\boldsymbol{r}) = \sum_j n_j \ln(|\boldsymbol{r} - \boldsymbol{r}_j|) \,. \tag{29}$$

Therefore, we rewrite the reduced Hamiltonian of Eq. 25 as

$$\mathcal{H} = -\frac{K}{2} \int d^2x [(\nabla\phi)^2 - 2\nabla\phi \cdot \nabla \times (\widehat{z}\psi) + (\nabla \times (\widehat{z}\psi))^2]. \tag{30}$$

Let us examine each of those terms separately. The first term will represents the spin-wave degree of freedom, a Gaussian part which can be integrated out. We have now isolated it from the rest, and will call it $\mathcal{H}_{sw}$. We treat the second term by integrating it by part. We get a surface term with $\phi$, which we assume vanishing at the boundary, and a divergence of a curl, making the integral of that second term disappear. The third term is simplified using $\nabla \times (\widehat{z}\psi) = -\widehat{z} \times \nabla\psi$. Therefore, we can write:

$$(\nabla \times (\widehat{z}\psi))^2 = (\nabla \times (\widehat{z}\psi)) \cdot (-\widehat{z} \times \nabla\psi) = -\widehat{z} \cdot (\nabla\psi \times (\nabla \times (\widehat{z}\psi))) = (\nabla\psi)^2. \tag{31}$$

We then write down the vortex part of our Hamiltonian, and perform an integration by part for which the surface part goes to 0 at infinity, and get

$$\mathcal{H}_v = -\frac{K}{2} \int d^2x \, (\nabla\psi)^2 = -\frac{K}{2}(\nabla\psi \cdot \psi)|_{S\to\infty} + \frac{K}{2} \int d^2x \, \psi\nabla\psi \tag{32}$$

$$= K\pi \sum_{i,j} n_i n_j \ln(|r_i - r_j|) \tag{33}$$

$$= \sum_i \mathcal{H}_{n_i}^{core} + 2K\pi \sum_{i<j} n_i n_j \ln(|r_i - r_j|). \tag{34}$$

In the last step, we have added the **core energy** of the vortices, which regulates the divergence of the other term when $i = j$. We then conclude with the following version of our partition function with $K = \beta J$ and $a$ the lattice spacing.

$$\mathcal{Z} = \int \mathcal{D}[\phi] \exp\left[-\frac{K}{2} \int d^2x \, (\nabla\phi)^2\right] \times \sum_{N=0}^{\infty} \frac{1}{(N!)^2} \int \left(\prod_{i=1}^{2N} \frac{d^2x_i}{a^2}\right) e^{\mathcal{H}_v} \tag{35}$$

$$= \mathcal{Z}_{S.W.} \mathcal{Z}_T \tag{36}$$

$$= \mathcal{Z}_{S.W.} \sum_{N=0}^{\infty} \frac{y_0^{2N}}{(N!)^2} \int \left(\prod_{i=1}^{2N} \frac{d^2x_i}{a^2}\right) \exp\left[2K\pi \sum_{i<j} n_i n_j \ln(|r_i - r_j|)\right]. \tag{37}$$

Let us explain how we went through those terms. The $1/(N!)^2$ factor comes from the combinatorial factor of having $N$ vortices; for any arrangement $\{r_i\}$ of $N$ vortices, there are $N!$ ways to identically arrange themselves. Furthermore, without loss of generality, we can set the whole configuration as neutral, such that vortices come in pair with the antivortices. We must then exclude $N!$ identical combinations of vortices, hence the $1/(N!)^2$ factor. Also, the integral over $d^2x_i$ is being modulated by a $1/a^2$ term that accounts for the lattice spacing.

Finally, we have a term $y_0 = \exp(\mathcal{H}_{\pm 1}^{core}) = \exp{-\beta E_c}$. This one comes from a constraint we have on the system: $\sum_i n_i = 0$, i.e. a neutrality condition. We can see this constraint arise when we consider a finite size system of size $L$, for which the boundary term we have so recklessly discarded actually gets bigger and bigger, with $\psi \to \ln L \sum_i n_i$. Then, imposing the neutrality condition explicitly takes care of this boundary term. We also see that any $n_i \neq \pm 1$ is going to be

entropically disfavored, as they make it harder for the system to satisfy the constraint. This is why we have removed the core energy $E_c$ of those unit charge vortices (note: the core energy does not depend on the sign of the vortex). This term $y_0$ is called the fugacity, and will be of crucial importance later. It is linked to the chemical potential of having a vortex in the system.

We then have our full partition function for the topological defects, the same as that for a neutral Coulomb plasma in two dimension [11]:

$$\mathcal{Z}_T = \sum_{N=0}^{\infty} \frac{y_0^{2N}}{(N!)^2} \int \left( \prod_{i=1}^{2N} \frac{d^2 x_i}{a^2} \right) \exp \left[ 2K\pi \sum_{i<j} n_i n_j \ln(|\mathbf{r}_i - \mathbf{r}_j|) \right], \tag{38}$$

which we have successfully separated from the Gaussian part of the partition function of the XY model, *i.e.* the one responsible for the spin-wave treatment of the 2D XY model. We see that this $\mathcal{Z}_{\text{S.W.}}$ is perfectly analytic, and therefore presents no finite temperature (finite $K$) phase transition. Indeed, there is no spontaneous continuous symmetry breaking, as we expect from a $O(2)$ model in 2D. Hence, we expect all of the phase transition properties to come from the topological term, $\mathcal{Z}_T$.

Nevertheless, we can consider independently the effect of the spin waves on renormalizing the coupling $K$. Starting from $\mathcal{Z}_{\text{S.W.}}$, we can add an anharmonic fluctuation in $\phi$ ($(\partial_x \theta)^4$ and $(\partial_y \theta)^4$).

$$\mathcal{Z}_{\text{S.W.}} = \int \mathcal{D}[\phi] \exp \left[ -\frac{K}{2} \int d^2 x \, (\nabla \phi)^2 + (\partial_x \theta)^4 + (\partial_y \theta)^4 \right]. \tag{39}$$

Doing a mean-field approximation on these anharmonic fluctuations, one writes $(\partial_x \theta)^4 \simeq \langle (\partial_x \theta)^2 \rangle (\partial_x \theta)^2$. Furthermore, via the isotropy of the system, we have $\langle (\partial_x \theta)^2 \rangle = \langle (\partial_y \theta)^2 \rangle$. Computing this average through the Gaussian integral leads to $\langle (\partial_x \theta)^2 \rangle = \frac{k_B T}{4J}$. Then, putting together the similar terms, one recovers the spin-wave quadratic Hamiltonian, but with a new $K$

$$K' = K(1 - \frac{k_B T}{4J}). \tag{40}$$

Indeed, spin waves renormalize slightly the coupling constant of the XY model. For $T_{KT} \sim \pi J/2$, which we derived earlier through the heuristic argument of Kosterlitz and Thouless, $K$ is still finite and large. The transition is due to the topological term. Note that this effect from the spin waves can be seen in numerical studies of the XY model.

## 3.2 RG flow equations

We now proceed to do a perturbative treatment of this interacting Hamiltonian in orders of the fugacity $y_0$. We will limit our expansion to second order, for which there are only two charges in the system. Those two charges will be at positions $\mathbf{s}$ and $\mathbf{s}'$, while we will study the screening of those two charges by the rest of the system in order to calculate the effective interaction between points $\mathbf{r}$ and $\mathbf{r}'$. For simplicity, the primed coordinates are negative charges ($n = -1$) and the unprimed are positive ($n = 1$). We have that our effective Hamiltonian $\mathcal{H}_{\text{eff}}(r - r') \simeq -2\pi K_{\text{eff}} \ln(r - r')$ is obtained from averaging the interaction between the external charges at $\mathbf{r}$ and $\mathbf{r}'$:

$$e^{\mathcal{H}_{\text{eff}}(r-r')} = \langle e^{-2K\pi \ln(|\mathbf{r}-\mathbf{r}'|)} \rangle_T . \tag{41}$$

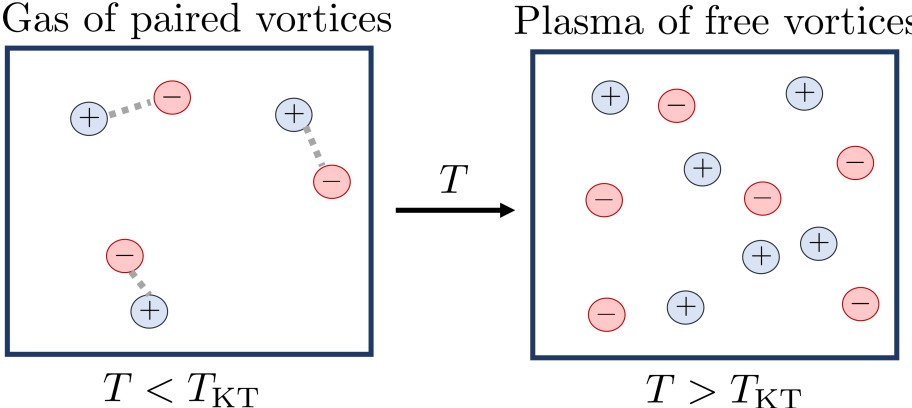

Figure 3: Schematic diagram showing deconfinement of the vortex pairs above the Kosterlitz-Thouless phase transition.

Performing this average is no easy calculation - we present it in its entirety at Appendix A. The end result is that

$$e^{\mathcal{H}_{\text{eff}}(r-r')} = e^{-2K\pi \ln(r-r')} e^{8K^2\pi^4 y_0^2 \ln(r-r') \int_1^\infty dx\, x^{3-2K\pi} + O(y_0^4)}, \tag{42}$$

with the expectation value taken with respect to $\mathcal{Z}_T$, the partition function of the topological degrees of freedom. The minus sign comes from $n_r * n_{r'} = 1 * (-1) = -1$ because of opposite charges. We can therefore write an equation concerning the effective interaction $K_{\text{eff}}$, the effective coupling constant after having integrated out the screened interaction of a pair of vortices.

$$K_{\text{eff}} = K - 4\pi^3 K^2 y_0^2 \int_1^\infty dx\, x^{3-2\pi K} + O(y_0^4). \tag{43}$$

We now proceed to get the renormalization group flow for the free parameters in the problem, the fugacity $y_0 = e^{-\beta E_c}$ with $E_c$ the core energy of a singular vortex, and the coupling constant $K$. We do a treatment that is akin to the one done by José, Kirkpatrick, Kadanoff and Nelson in 1977 [6,12]. This procedure is, again, presented in detail in Appendix **??**. The result is that

$$\frac{dK^{-1}}{dl} = 4\pi^3 y_0^2 + O(y_0^4),$$
$$\frac{dy_0}{dl} = (2 - \pi K)y_0 + O(y_0^3), \tag{44}$$

where $l$ is a lengthscale over which we view our system. At $l = a$ the system is described by the bare parameters $K = J/T$ and $E_c = J\pi^2/2$ [13]. As $l$ is increased, the parameters flow to their fixed point values. In the following subsection, we will analyze the RG flow in detail.

## 3.3 Nelson-Kostelitz jump

What does the RG flow for the XY model, as written in Eq. 44, tell us? Let us focus first on the two temperature extremes. In the **low-temperature limit**, *i.e.* when $K^{-1}$ is small, the flow of $y_0$

reduces it until it terminates on a line of fixed points at which $y_0 = 0$ and $K_{eff}^{-1} \leq \pi/2$. In this regime, the fugacity becomes irrelevant. This is an insulating phase of the 2D Coulomb gas, where there are no free charges, as can be seen on the left of Fig. 3. In the XY model reference frame, this means that in this regime, vortices, if they exist, are tightly bound together in pairs with some finite radius. Because we defined $y_0 = \exp(-\beta E_c)$ with the core energy, the renormalization of the fugacity in the low temperature limit leads to an infinite core energy. Since $y_0$ represent the chemical potential of free vortices, we have rigorous proof that below a certain temperature $T_{KT} = \pi J/2$, there exists a phase for which *only* bound vortex-antivortex pairs can exist, and these pairs become less and less numerous as temperature increase due to the diverging renormalized core energy. As we have seen before, this phase displays quasi-long-range order (QLRO).

In the **high-temperature limit**, the RG flow is in a runoff regime, which leads to ever increasing values of $K^{-1}$ and $y_0$. This implies an eventual breakdown of the perturbation theory. With $y_0$ ever increasing, this means that it becomes more and more favorable for free vortices to exist. This is the metallic regime of the 2D Coulomb gas, where free charges abound, and correlation functions decay exponentially. The presence of free vortices (free charges) kills the QLRO. The resulting plasma of free vortices is showed on the right of Fig. 3.

In order to simplify our analysis of the RG and understand better the phase transition, we fix our attention to the region close to the fixed point ($K_C^{-1} = \pi/2, y_0 = 0$). As can be seen in Eq. 44, at this point, the beta functions are null and the parameters are scale invariant. This fixed point defines the critical temperature $T_{KT}$, such that

$$\frac{J(T_{KT})}{T_{KT}} = \frac{2}{\pi}, \tag{45}$$

where we brought back $K = J/T$. When the renormalized coupling $J$ at a given temperature satisfies this equation, that is $T_{KT}$ the Kosterlitz-Thouless phase transitions. Let us now delve our attention close to this point. Setting $t = K^{-1} - \pi/2$ and $y = y_0$, one gets the following non-linear flow equations near the fixed point.

$$\frac{dt}{dl} = 4\pi^3 y^2 + O(ty^2, y^4),$$
$$\frac{dy}{dl} = \frac{4}{\pi} ty + O(t^2 y, y^3). \tag{46}$$

The resulting flow for these equations is represented in figure 4. We can see three very distinct regions, the structure of which is uncovered when we see that the equations 46 have that they preserve the quantity $c = t^2 - \pi^4 y^2$, such that $dc/dl = 0$. This means that around the fixed point, we have a series of hyperbolae with asymptotic limits $y = \pm t/\pi^2$. These are in black in the figure, and correspond to $c = 0$, the critical trajectories.

The three different regions are as follows:

- $c < 0$ : These trajectories are above the critical one, and their flow goes from large $y$ and small $K^{-1}$, to small $y$, and then again to large $y$. These curves are the upper region in figure 4.

- $c > 0$ and $t < 0$: This is the low-temperature regime in which flows terminate on the line of fixed points with $y = 0$.

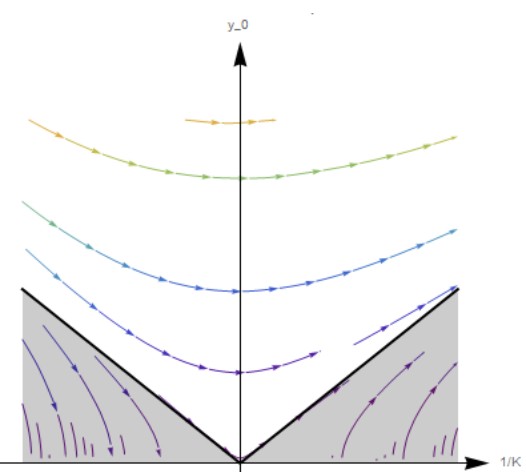

Figure 4: The RG-flow around the fixed point, with the characteristic hyperbolae $c = (K^{-1} - \pi/2)^2 - \pi^4 y_0^2$. The shaded regions have $c > 0$.

- $c > 0$ and $t > 0$: Here, flows start from $y = 0$ but the fugacity is renormalized to infinity. These are unphysical.

The critical line itself separates trajectories that flow to $y = 0$ from those that flow to $y \to \infty$, and on it, we have $t_c = -\pi^2 y_c^2$. Hence, a small but finite fugacity renormalizes the critical temperature to $K_C^{-1} = \pi/2 - \pi^2 y_0$.

We can now compare the value of the renormalized coupling constant $K(b, T)$ slightly above and slightly below the transition temperature $T_{KT}$, we see that, as the $b \to \infty$, we have

$$\lim_{\Delta T \to 0} \lim_{b \to \infty} [K(b, T_{KT} - \Delta T) - K(b, T_{KT} + \Delta T)] = \frac{2}{\pi} . \tag{47}$$

Hence, there is a jump in the renormalized coupling at $T = T_{KT}$. Indeed, $\forall T > T_{KT}$, the RG flow tell us that $K^{-1}$ will flow to infinity, hence $K_{\text{eff}} \to 0$. We then have:

$$J_{\text{eff}}(T_{KT}) = \frac{2}{\pi} T_{KT} . \tag{48}$$

This discontinuous jump of the renormalized coupling $J_{\text{eff}}$ is called the Nelson-Kosterlitz universal jump. In the thermodynamic limit, the temperatures below $T_c$ have a finite renormalized value of $J_{\text{eff}}$. At higher temperatures, it renormalizes to 0, hence there should be a jump, a discontinuity, at $T = T_{KT}$. In the Monte-Carlo section, we will show how we can measure this effective coupling constant, which we will call the spin stiffness. The Nelson-Kosterlitz jump then becomes a criteria we can use to pinpoint the location of the Kosterlitz-Thouless phase transiton with accuracy.

Furthermore, the Nelson-Kosterlitz jump has been experimentally confirmed in 2D planar systems such as thin films of Helium-4. The superfluid density corresponds to this effective coupling $J_{\text{eff}}$ and is seen jumping discontinuously to a finite value at the critical temperature.

## 3.4 Correlation Length and Specific Heat

In the high-temperature limit, one can also linearize the RG flow equations. In this phase, we have $c = t^2 - \pi^4 y^2 > 0$. We can then write it as $c = b^2(\frac{T - T_{KT}}{T_{KT}})$, and then integrate fully the recursion relation $dt/dl = 4\pi^3 y^2 = 4/\pi(t^2 + b^2(\frac{T - T_{KT}}{T_{KT}}))$. We find:

$$\frac{4l}{\pi} \simeq \frac{1}{b}\sqrt{\frac{T_{KT}}{T - T_{KT}}} \arctan\left(\frac{t}{b}\sqrt{\frac{T_{KT}}{T - T_{KT}}}\right). \tag{49}$$

We limit the integration when $t(l) \sim y(l) \sim 1$, because beyond that, our approximations are invalid and we enter a regime in which the pertubation scheme is void. Using the fact that $b$ is very large and we take $|T - T_c| \ll 1$, then $\arctan(\frac{1}{b}\sqrt{\frac{T_{KT}}{T - T_{KT}}}) \approx \pi/2$. Therefore, we have that the value $l^*$ at which we stop integrating is given by $l^* = \frac{\pi^2}{8b}\sqrt{\frac{T_{KT}}{T - T_{KT}}}$. The correlation length is given by $\xi \approx ae^{l^*}$, which gives:

$$\xi \approx a \exp\left[\frac{\pi^2}{8b}\sqrt{\frac{T_{KT}}{T - T_{KT}}}\right]. \tag{50}$$

This expression clearly diverges as $T \to T_{KT}$, but not in the same way as usual second-order phase transitions, in which the divergence is as a power law. This is a clear signal that the phase transition at $T_{KT}$ defies the normal expectations at thermal phase transitions.

Furthermore, for low-temperatures, a similar analysis reveals that $\xi = \infty$ at all temperatures below $T_{KT}$. The XY model is critical in its low-temperature phase, thus the quasi-long-range order and the algebraic correlations.

Finally, a small comment on the specific heat, which has a peculiar behavior in the XY model. One can look first at the singular part of the free energy when approaching the phase transition from the high-temperature regime, and finds that

$$f_{sing} \propto \xi^{-2} \propto \exp\left(-\frac{\pi^2}{4b}\sqrt{\frac{T_{KT}}{T - T_{KT}}}\right). \tag{51}$$

This free energy only has an essential singularity, in the sense that it is singular, but all derivatives exist and are finite at $T \neq T_{KT}$. Hence, one expects the heat capacity to be smooth at the transition. Furthermore, solving the RG equations as well as numerical analysis show a maximum at a temperature higher than $T_{KT}$ associated with the ramping up of the production of vortices (not with their unbinding). This round shoulder is usually located at $T_{cv} \sim 1.1\ T_{KT}$ in Monte-Carlo studies - the prefactor is not universal. This behavior is shown in Fig. 5.

## 4 Monte-Carlo approach

Consider an observable $O$, which could be the energy, magnetization or anything else. One could measure such an observable for a set configuration $\mathcal{C}_a = \{\theta_i\}$ and obtain $O(a)$. In statistical mechanics and condensed matter, our goal is not to simply compute this value once, but rather to compute expectation values

$$\langle O \rangle = \frac{1}{\mathcal{Z}} \text{tr}\{Oe^{-\beta H}\}, \tag{52}$$

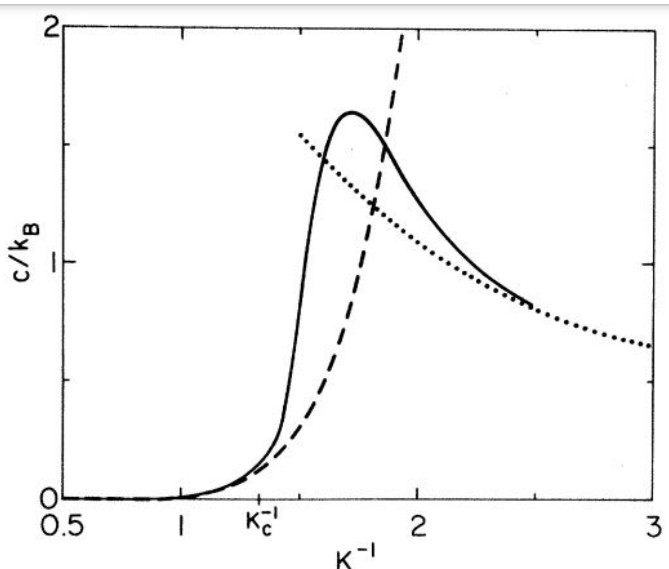

Figure 5: Specific heat behavior due to the vortex excitations (full curve) obtained from the RG equations. Horizontal axis is $K^{-1} = T/J$. Dashed lines at high and low-temperature expansion results. At the critical temperature no singular behavior occurs. Taken from Ref. [14].

where $\mathcal{Z} = \text{tr}\{e^{-\beta H}\}$ is the partition function, and the trace is taken over all the states of the system for a given temperature. This would be easy if we knew how to write **all** possible configurations in a succint way, with their associated energies and observables. However, for most cases, we need to resort to numerical tools to estimate this average.

The key insight into the Monte-Carlo method is to harness the power of random updates to the configurations. The set of such random updates is called a Markov chain. This was first invented by physicist Stanislaw Ulam while working for the Manhattan project - he realized that while exact conventional mathematical methods were unable to solve a problem concerning neutron diffusion, a random approach might work much more efficiently and faster. The name Monte-Carlo came from a colleague, Nicholas Metropolis, who would later propose a class of local updates which we will see below, who likened the random aspect to the games of chance preformed at the famous Monaco casino.

If we were to simply propose a large number of configurations $\mathcal{C}_a$, and then calculate the observable average with the weight fact $\exp(-E_a/k_B T)$ we would be considering states which are, by default, "improbable" for this specific temperature. For example, if $T$ is high, maybe you proposed a very ordered state with a small entropy and low energy, which should be unlikely to occur. Therefore, one needs to guide the random process, and random moves need to be accepted on some conditions. That way, we do not waste our time on unrepresentative samples, but instead generate a new, improved probability distribution based on the frequency of certain configurations.

Consider a configuration $\mathcal{C}_a$. A successive, updated state $\mathcal{C}_{a+1}$ is constructed from the previous state through a transition probability $W(\mathcal{C}_a \rightarrow \mathcal{C}_1)$. There is a strict requirement on this transition

probability, and that is called detailed balance:

$$\frac{W(\mathcal{C}_a \to \mathcal{C}_1)}{W(\mathcal{C}_{a+1} \to \mathcal{C}_a)} = \frac{P_a}{P_{a+1}}, \tag{53}$$

with $P_a$ the probability to encounter the state configuration $a$. Detailed balance is very simply a statement that in equilibrium, the transfer from $a$ to $a+1$ is reversible, i.e. equal to its opposite from $a+1$ to $a$. For a system in thermodynamical equilibrium, the probabilities $P_a$ are simply Boltzmann factors, such that if configuration $\mathcal{C}_a$ has energy $E_a$, then $P_a = \exp(-E_a/k_B T)$. We can extract a relationship between the transition probabilities $W$, but it is not unique in general.

Another key element of the Monte-Carlo method is for the Markov chain to be ergodic, *i.e.* that through this random walk in the spce of configurations, **all** possible configurations could be reached. It does not mean they all need to be reached proportionally; in fact, quite the opposite! But a finite series of random updates on $\mathcal{C}_a$ needs to access ant $\mathcal{C}_b$.

Therefore, an Monte-Carlo algorithm that respects **detailed balance** and **ergodicity** is valid. Some may be more efficient then others. In the following subsections, we will present two types of algorithms for the XY model: the simple local Metropolis update, and the glogal Wolff cluster update.

Once an algorithm is constructed, averages of observables $O$ become quite simple to evalue, since the directed random walk in configuration space returns some configuration in a manner proportional to their Boltzmann weight. We then have

$$\langle O \rangle = \frac{1}{M} \sum_a O(a), \tag{54}$$

where $M$ is the number of configurations $\mathcal{C}_a$ obtained by the Monte-Carlo process.

## 4.1 Local updates: Metropolis

This is the type of update first proposed by Metropolis, Rosenbluth, Rosenbluth and Teller in 1953 [15]. It is local, whereby a site $i$ on the square lattice is chosen at random, and then its angular value $\theta_i$ is updated to a new $\theta_i'$. The proposed update is either accepted or not. The full procedure is

1. Start with a random configuration $\mathcal{C}_a$;

2. Randomly choose a site $i$ and generate a random angle $\phi$;

3. Calculate the energy change $\Delta E = E_a - E_a'$, where $E_a$ is the energy of configuration $a$ and $E_a'$ is the energy for a new configuration $\mathcal{C}_a'$ where all angles are identical except $\theta_i \to \phi$.

4. Consider where to accept or reject this update:

   (a) **If $\Delta E < 0$**, accept the change for spin $i$.

   (b) **If $\Delta E \geq 0$**, accept the change with probability $p_c = e^{-\Delta E/T[m]}$. This is done by choosing a random number $p \in [0, 1]$, and checking whether $p < p_c$.

   (c) **If $p > p_c$**, reject the change, so that $\mathcal{C}_a' = \mathcal{C}_a$.

5. Whether or not the update has been accepted, call the new configuration $\mathcal{C}_{a+1} = \mathcal{C}_a'$.

6. Go back to step 2. Repeat $M$ times.

Because interactions in the XY model are only on nearest neighbors, a direct expression for $\Delta E$ can be obtained:

$$\Delta E = -J \sum_{\boldsymbol{\mu}} [\cos(\theta_i - \theta_{i+\boldsymbol{\mu}}) - \cos(\phi - \theta_{i+\boldsymbol{\mu}})] , \tag{55}$$

where $\boldsymbol{\mu} = \pm \hat{x}, \pm \hat{y}$ are the four different directions to nearest neighbors on the square lattice.

For very large lattices, this is a very long process, as we will need $M \sim N = L^2$ updates to even have possibly updated all angles on the lattice at least once. This problem is remedied by the next type of update, which we will favor in our numerical studied of the XY model.

## 4.2 Global updates: Wolff

The philosophy of the Wolff algorithm [16] is to construct one large cluster of quasi-parallel spins, that are each added to the cluster with a probability $P_{\text{add}}$. Then, we flip the whole cluster. Ergodicity in such a method is respected because there is always a non zero probability that a cluster will consist of only the initial spin, that would then be flipped. In that limit, it becomes a local Metropolis update which is ergodic. For an Ising model with $\mathbb{Z}_2$ variables $\sigma_i = \pm 1$, such a cluster creation is simple, as only parallel spins can be added. In the case of XY models with $O(2)$ symmetry, one needs to be cleverer.

A general flipping operation is introduced. For a given random vector $\boldsymbol{r} \in O(2)$, such that $\boldsymbol{r} = (\cos\phi, \sin\phi)$, the reflection of a spin $\boldsymbol{\sigma}_i = (\cos\theta_i, \sin\theta_i)$ is described with respect to the hyperplane orthogonal to $\boldsymbol{r}$:

$$R(\boldsymbol{r})\boldsymbol{\sigma}_i = \boldsymbol{\sigma}_i - 2(\boldsymbol{\sigma}_i \cdot \boldsymbol{r})\boldsymbol{r} . \tag{56}$$

This is an idempotent operation ($R(\boldsymbol{r})^2 = 1$) and it is invariant under global rotation ($R(\boldsymbol{r})\boldsymbol{\sigma}_i)\cdot(R(\boldsymbol{r})\boldsymbol{\sigma}_j) = \boldsymbol{\sigma}_i\cdot\boldsymbol{\sigma}_j$. Using this definition of flipping, we can describe a Monte-Carlo algorithm as

1. Start with a random configuration $\mathcal{C}_a$;

2. Generate a random angle $\phi$, which results in a random vector $\boldsymbol{r} = (\cos\phi, \sin\phi)$. Initiate an empty list of sites which is the cluster $\mathbb{C}$;

3. Randomly choose a site $i$ and add site $i$ as the first site in cluster $\mathbb{C}$;

4. Pick a site from cluster $\mathbb{C}$ which has not yet been updated.

5. Flip $\boldsymbol{\sigma}_i$, such that $\boldsymbol{\sigma}'_i = R(\boldsymbol{r})\boldsymbol{\sigma}_i$.

6. **For** all neighbours $j = i \pm \boldsymbol{\mu}$ to site $i$:

    (a) Calculate $\Delta E_{\langle i,j \rangle} = \boldsymbol{\sigma}'_i \cdot [1 - R(\boldsymbol{r})]\boldsymbol{\sigma}_j$.

    (b) **If** site $i$ is already in cluster $\mathbb{C}$, then do nothing.

    (c) **If** $\Delta E_{\langle i,j \rangle} > 0$, then do nothing.

    (d) **If** $\Delta E_{\langle i,j \rangle} < 0$, choose a random number $p \in [0,1]$, and define $p_c = 1 - \exp(\Delta E_{\langle i,j \rangle}/)$.

    (e) **If** $p < p_c$, add site $\boldsymbol{\sigma}_j$ to the cluster $\mathbb{C}$.

    (f) **Else**, for $p > p_c$, do nothing with site $j$.

7. Go back to step (4) until all sites from the cluster $\mathbb{C}$ have been tested (i.e. all their neighbors have been proposed as add-ins). Once there are no new sites to be considered, call the new configuration $\mathcal{C}_{a+1} = \mathcal{C}'_a$, where all sites that were accepted in the cluster have been flipped.

8. Go back to step 2. Repeat $M$ times.

Note that for this definition of the spin flipping procedure $R(\boldsymbol{r})$, we have $\boldsymbol{\sigma}'_i \cdot [1-R(\boldsymbol{r})]\boldsymbol{\sigma}_j = 2(\boldsymbol{\sigma}'_i \cdot \boldsymbol{r})(\boldsymbol{\sigma}_j \cdot \boldsymbol{r})$. Incorporating the energy comparison steps together, the probability of accepting a link can be more succinctly written as

$$p_{\langle i,j \rangle} = 1 - \exp\left(\min\left[0, 2(\boldsymbol{\sigma}'_i \cdot \boldsymbol{r})(\boldsymbol{\sigma}_j \cdot \boldsymbol{r})/T\right]\right). \tag{57}$$

This algorithm can be generalized to models with general $O(N)$ continuous symmetry. We can see that a single Wolff update has the potential of flipping **all** sites on the lattice (it will in fact do that as $T \to 0$). Therefore, we need to do less Monte-Carlo moves to update the whole system than in the Metropolis method.

We now examine how this algorithm respects detailed balance by considering two configurations, $\mathcal{C}_a$ and $\mathcal{C}'_a$, different by a flip $R(\boldsymbol{r})$ on a cluster $\mathbb{C}$. We wish to justify the particular choice of probability to add a site or not.

The condition of detailed balance is that the ratio of the probability to go from one configuration $\mathcal{C}_a$ to the other and then reverse this process is equal to the ratio of the Boltzmann weight. This is expressed as

$$\frac{W(\mathcal{C}_a \to \mathcal{C}'_a)}{W(\mathcal{C}'_a \to \mathcal{C}_a)} = e^{-\beta(E_a - E'_a)}, \tag{58}$$

with $\beta = 1/T$. Then, we write the following steps, with the notation that $\partial\mathbb{C}$ corresponds to the boundary of the cluster $\mathbb{C}$.

$$\frac{W(\mathcal{C}_a \to \mathcal{C}'_a)}{W(\mathcal{C}'_a \to \mathcal{C}_a)} = \prod_{\langle i,j \rangle \in \partial\mathbb{C}} \frac{1 - W(R(\boldsymbol{r})\boldsymbol{\sigma}_i, \boldsymbol{\sigma}_j)}{1 - W(R(\boldsymbol{r})\boldsymbol{\sigma}_i, \boldsymbol{\sigma}_j)} \tag{59}$$

$$= \prod_{\langle i,j \rangle \in \partial\mathbb{C}} \frac{e^{-\beta[R(\boldsymbol{r})\boldsymbol{\sigma}_i \cdot \boldsymbol{\sigma}_j]}}{e^{-\beta[R(\boldsymbol{r})\boldsymbol{\sigma}'_i \cdot \boldsymbol{\sigma}'_j]}} \tag{60}$$

$$= \exp\left\{\beta \sum_{\langle i,j \rangle \in \partial\mathbb{C}} \boldsymbol{\sigma}_i \cdot [R(\boldsymbol{r})-1]\boldsymbol{\sigma}_j\right\} \tag{61}$$

$$= \exp\left\{\beta \sum_{\langle i,j \rangle} \boldsymbol{\sigma}'_i \cdot \boldsymbol{\sigma}'_j - \boldsymbol{\sigma}_i \cdot \boldsymbol{\sigma}_j\right\} \tag{62}$$

The first step is because if a spin $i$ is within a cluster $\mathbb{C}$ but not on the edge $i \notin \partial\mathbb{C}$, then the flipping of a complete cluster will have no energy cost associated with these interior spins $i$, as they are all flipped by the same amount, which conserves the energy. The only change in energy from flipping a cluster comes from the edge contributions.

The transition probabilities here are given by the probability that all bonds $\langle i,j \rangle$ for $i \in \mathbb{C}$ and $j$ crossing the edge are not added to the cluster (*i.e.* the cluster stopped growing). That is the product of $P(\text{not add}) = 1 - W(R(\boldsymbol{r})\boldsymbol{\sigma}_i, \boldsymbol{\sigma}_j) = \exp(-\beta R(\boldsymbol{r})\boldsymbol{\sigma}_i \cdot \boldsymbol{\sigma}_j)$ (all bonds crossing the edge

that have $\Delta E > 0$ will lead to a $\times 1$ in the product so they won't be addressed). It also makes use of the fact that $R$ is an idempotent operation, *i.e.* if it is applied twice, it returns the same state. The last equality uses again the fact that sites in the bulk of the cluster cost zero energy to reflect. This choice of bond-adding probability then respects detailed balance.

We note however that the detailed balance is not satisfied when adding individual sites to the cluster, because spins that are added are always flipped, but once the cluster is completely built, detailed balance is respected globally, via the rejection of spins along the edge.

Another version of this algorithm can be more efficiently written using the following definition for the spin flip. Instead of randomly selecting a vector $\boldsymbol{r}$ and flipping the parallel part of the spin with respect to the vector $\boldsymbol{r}$, we simply randomly select an angle $\phi \in [0, \pi)$, and we have that the rotation function for a spin $\boldsymbol{\sigma}_i = (\cos \theta_i, \sin \theta_i)$ is:

$$R_\phi \boldsymbol{\sigma}_i = (\cos(2\phi - \theta_i), \sin(2\phi - \theta_i)), \tag{63}$$

so that $\theta_i \rightarrow \theta'_i = 2\phi - \theta_i$. One sees via simple geometry that this is the same effect as the previous definition for $R(\boldsymbol{r})$. We also see that the idempotent property is preserved, having $R_\phi^2 = 1$, and also

$$((R_\phi \boldsymbol{\sigma}_i) \cdot (R_\phi \boldsymbol{\sigma}_j) = \cos((2\phi - \theta_i) - (2\phi - \theta_j)) = \cos(\theta_i - \theta_j) = \boldsymbol{\sigma}_i \cdot \boldsymbol{\sigma}_j. \tag{64}$$

Hence, the second property is also respected. Using this definition, our energy comparison criteria for accepting or rejecting a link becomes

$$\Delta E_{\langle i,j \rangle} = -J[\cos(\theta_i - \theta_j) - \cos(\theta_i + \theta_j - 2\phi)]. \tag{65}$$

This method shows a significant increase in speed, due to the fact that a lot fewer operations are being done in order to flip a spin - it is simply a matter of substraction and multiplication, with no dot products. Finally, note that there exists many other global updates that are also efficient, such as the Swendsen-Wang algorithm [17], which is another global update, Heat-Bath overrelaxation methods [18–20], or the now nearly ubiquitous Worm algorithm. This last one, created by Prokofiev and Svistunov [21] in 2001, while challenging to implement, is actually one of the fastest and most efficient algorithm to simulate the XY model. It necessitates great care in setting up, as one works in a dual, current space, where updates are proposed.

## 4.3 Observables and spin stiffness

The following quantities are readily understood as measurements of the energy per site, the specific heat per site, the magnetization per site and the susceptibility per site:

$$e = \frac{E}{N} = -\frac{J}{N}\left\langle \sum_{\langle i,j \rangle} \cos(\theta_i - \theta_j) \right\rangle, \qquad c_V = \frac{\langle E \rangle^2 - \langle E^2 \rangle}{NT^2},$$

$$m = \frac{|\boldsymbol{M}|}{N} = \frac{1}{N}\left\langle |\sum_i \{\cos \theta_i, \sin \theta_i\}| \right\rangle, \qquad \chi = \frac{\langle |\boldsymbol{M}|^2 \rangle - \langle |\boldsymbol{M}| \rangle^2}{NT}. \tag{66}$$

At any finite temperature, $|\boldsymbol{M}|$ will go to zero in the thermodynamic limit ($L \rightarrow \infty$). However, since the low-temperature is quasi-long-range-ordered, it will go to zero *algebraically* there,

whereas it will fall to 0 exponentially in the high-temperature phase. For finite system sizes, we can still observe finite valued $m$ at low-temperature.

Furthermore, we can calculate explicitly the presence of topological defects in the angle $\theta$ around an elementary square plaquette $\square$. The vorticity around a given plaquette is defined as

$$\omega_{v,\square} = \frac{1}{2\pi} \sum_{\langle ij \rangle \in \square} \nabla \theta_{ij} \,, \tag{67}$$

where the sum runds over all the bonds on the given plaquette in a counterclockwise was, and $\nabla \theta_i j$ is defined in the interval $[-\pi, \pi]$. Therefore, it is equal to $\nabla \theta_i j = \theta_i - \theta_j - 2\pi$ if $\theta_i - \theta_j > \pi$ and to $\nabla \theta_i j = \theta_i - \theta_j + 2\pi$ if $\theta_i - \theta_j < -\pi$. The total vortex-antivortex pair density can be described as the sum over all plaquettes in the system, counting the absolute value of the vorticity:

$$\omega_v = \frac{1}{2N} \left\langle \sum_{\square} |\omega_{v,\square}| \right\rangle \,. \tag{68}$$

This is another important indicator of the physics at play in the classical 2D XY model.

There is another essential quantity we wish to measure for the XY model: the spin stiffness. Let us first motivate what we seek to measure here. We showed in earlier sections that the long-wavelength, continuum limit of the XY model has the Hamiltonian $H = \frac{J_{\text{eff}}}{2} \int d\boldsymbol{x} (\nabla \theta(\boldsymbol{x}))^2$. We wish to extract a measurement that would be on the order of $J_{\text{eff}}$. If we were to apply a potential $\boldsymbol{A}$ on this model, we would obtain

$$H[\boldsymbol{A}] = -\frac{J_{\text{eff}}}{2} \int d^2 r |\nabla \theta(\boldsymbol{r}) - \boldsymbol{A}|^2 \,, \tag{69}$$

and we can express a generalized current [22] as

$$I_s^x[\boldsymbol{A}] = \frac{\delta \langle H[\boldsymbol{A}] \rangle}{\delta A_x(\boldsymbol{x})} = -J_{\text{eff}}[\nabla \theta(\boldsymbol{r}) - \boldsymbol{A}]_x \,, \tag{70}$$

$$\frac{\delta I_s^x[\boldsymbol{A}]}{\delta A_x(\boldsymbol{x})} = \frac{\delta^2 \langle H[\boldsymbol{A}] \rangle}{\delta A_x(\boldsymbol{x}) \delta A_x(\boldsymbol{x})} = J_{\text{eff}} \,. \tag{71}$$

Therefore, measuring the response of the system to a current in the limit of no current measures the effective interaction. This is related to the notion of generalized rigidity [22] - if $J_{\text{eff}}$ is 0, the system is rigid, i.e. will not respond to any "twists" or applied current. If $J_{\text{eff}} \neq 0$, the system will respond.

For the XY model on a square lattice, a similar contruction can be done. Assume we work with open boundary conditions. We can apply a twist $\phi$ uniformly through the system such that, at the $x$ boundary, $\theta_{L+1,j} = \theta_{1,j} + \phi$. This can be implemented as

$$\boldsymbol{\sigma}_i \to \{\cos(\theta_i + \phi(i_x - 1)/L), \sin(\theta_i + \phi(i_x - 1)/L)\} \,. \tag{72}$$

Then, the Hamiltonian with a twist becomes

$$H_{twist}[\phi] = -J \sum_{\langle i,j \rangle} \cos\left(\theta_j - \theta_i - \frac{\phi}{L}\widehat{\boldsymbol{e}}_{i,j} \cdot \widehat{\boldsymbol{x}}\right) \tag{73}$$

$$= -J \sum_{i,\boldsymbol{\mu}} \cos\left(\theta_{i+\boldsymbol{\mu}} - \theta_i - \frac{\phi}{L}\boldsymbol{\mu} \cdot \widehat{\boldsymbol{x}}\right) \,, \tag{74}$$

where $\mu = \pm\hat{x}, \pm\hat{y}$. We can rewrite this using Ampère's law and a vector field $A$ such that $A = \nabla\phi$. We then have:

$$H[A] = -J \sum_{i,\eta} \cos\left(\theta_{i+\eta} - \theta_i + \int_j^{j+\eta\cdot\hat{x}} A \cdot dl\right). \tag{75}$$

We have that, at $T = 0$, the energy of the system is just the Hamiltonian. We then look at the dimensions of the energy difference between a twisted-boundary system and a usual one. At leading order in $A$, this difference is quadratic in the applied field, and the constant that comes in are $\rho_s$, the spin stiffness and $N$ the number of sites (by dimensional analysis). We get $H[A] - H[0] = \rho_s A^2 N)/2$

From which we get the following definition of the spin stiffness at $T = 0$:

$$\rho_s = \frac{1}{N} \lim_{A\to 0} \frac{H[A] - H[0]}{A^2} = \frac{1}{N} \frac{\partial^2 H[A]}{\partial A^2}\bigg|_{A\to 0}. \tag{76}$$

From the examination of $H[A]$, we have that, at $T = 0$ when all spins are parallel $\theta_i = \theta_0$, then $\rho_s = J$. At finite temperature, entropic effects mean that we need to look at how the free energy changes with the twist field applied to the boundary conditions. We have the following definition:

$$\rho_s = \frac{1}{V} \frac{\partial^2 F[\phi]}{\partial \phi^2}\bigg|_{\phi\to 0} \qquad F[\phi] = -\frac{1}{\beta}\ln(\mathcal{Z}[\phi]), \tag{77}$$

with the partition function being the usual $\mathcal{Z}[\phi] = \int \mathcal{D}[\theta]e^{-\beta H[\phi]}$. Then, we write the Hamiltonian from 75 in a more simplified form:

$$H[\phi] = -J \sum_{\langle i,j\rangle_x} \cos(\phi + \theta_j - \theta_i) - J \sum_{\langle i,j\rangle_y} \cos(\theta_j - \theta_i). \tag{78}$$

We then need to use the full strength of the fact that $\phi$ is very small and will be taken to 0 later. Then, we have, to second order in $\phi$:

$$\cos(\phi + \theta_j - \theta_i) = \cos(\theta_j - \theta_i)\cos\phi - \sin(\theta_j - \theta_i)\sin\phi \tag{79}$$

$$= \cos(\theta_j - \theta_i)(1 - \frac{\phi^2}{2}) - \sin(\theta_j - \theta_i)\phi + O(\phi^3). \tag{80}$$

We can then rewrite our $\phi$-dependent Hamiltonian as:

$$H[\phi] = H[0] + \frac{\phi^2}{2}H_x - \phi I_x + O(\phi^3), \tag{81}$$

$$\text{with} \quad H_x = J \sum_{\langle i,j\rangle_x} \cos(\theta_j - \theta_i), \tag{82}$$

$$\text{and} \quad I_x = J \sum_{\langle i,j\rangle_x} \sin(\theta_j - \theta_i). \tag{83}$$

Here, $I_x$ is the generalized spin current associated with the generalized rigidity $\rho_s$, the spin stiffness. As one applies a twist to the boundary condition, there is a flux of angular momentum in the system, and then there is a torque felt at the two boundaries that are relatively twisted. The partition function can then be rewritten as (neglecting the $O(\phi^3)$ terms, notably in the series expansion of the exponentials):

$$
\begin{aligned}
\mathcal{Z}[\phi] &= \int \mathcal{D}[\theta] e^{-\beta H[\phi]} \\
&= \int \mathcal{D}[\theta] e^{-\beta H[0]} e^{-\beta \phi^2 H_x/2} e^{-\beta \phi I_x} \\
&= \int \mathcal{D}[\theta] e^{-\beta H[0]} (1 - \frac{\beta \phi^2 H_x}{2} + \cdots)(1 + \beta \phi I_x + \frac{1}{2}(\beta \phi I_x)^2 + \cdots) \\
&= \mathcal{Z}[0](1 - \frac{1}{2}\beta \phi^2 \langle H_x \rangle + \beta \phi \langle I_x \rangle + \frac{1}{2}\beta^2 \phi^2 \langle I_x^2 \rangle).
\end{aligned}
\tag{84}
$$

By symmetry, we have that $\langle I_x \rangle = 0$ in the ground state, where the calculation is done. Finally, we have that $\ln(1-x) \approx x$ when $x$ is small. Hence, we can write the free energy as:

$$
F[\phi] = F[0] + \frac{1}{2}\phi^2 (\langle H_x \rangle - \beta \langle I_x^2 \rangle),
\tag{85}
$$

which, using the definition provided by equation 77, gives

$$
\rho_s = \frac{1}{N}(\langle H_x \rangle - \beta \langle I_x^2 \rangle)
\tag{86}
$$

$$
= \frac{1}{Nd}(\langle H \rangle - \beta \sum_{\alpha=1}^{d} \langle I_\alpha^2 \rangle).
\tag{87}
$$

The last line is a generalization for a $d$-dimensional system, with the sum taken over all the different directions possible on the lattice. This is only true for an isotropic system. In the presence of anisotropic interactions, $d$ different stiffness parameters need to be included. The spin stiffness for the XY model is then

$$
\rho_s = \frac{J}{N}\left\langle \sum_{\langle i,j \rangle} \cos(\theta_j - \theta_i)(\widehat{e}_{i,j} \cdot \widehat{x})^2 \right\rangle - \frac{J^2}{TN}\left\langle \left\{ \sum_{\langle i,j \rangle} \sin(\theta_j - \theta_i)\widehat{e}_{i,j} \cdot \widehat{x} \right\}^2 \right\rangle,
\tag{88}
$$

$\widehat{e}_{i,j}$ is a unit vector from site $i$ to site $j$, and the twist is applied in the $x$ direction. We could have done it in the $y$ direction, but the system is isotropic so it really does not matter in which direction the measurement is taken. We have a new observable, $\rho_s$, which can be measured. There is no need to actually apply twisted boundary conditions - this formula can be calculated though the Monte-Carlo process.

Note that, as low temperature well below the KT transition, spin waves may still proliferate in the system. Since these degrees of freedom are Gaussian, their contribution to the spin stiffness can be calculated exactly, and this leads the behavior $\rho_s \sim J(1 = \frac{T}{4J} + \cdots)$. We will show that this contribution readily be seen in Monte-Carlo data.

## 4.4   Temperature sweep

Now that we know how to go from one configuration to another well-selected configuration, and we know what to measure, we have the task of setting up a full Monte-Carlo routine. In its simplest form, this involves an annealing process, where at each temperatures, a series of Monte-Carlo steps are done first to thermalize the system, then to obtain measurements. The thermalization step is crucial, as otherwise, the generated configurations may not truly represent the temperature the simulation is set at, *i.e.* they are not at thermodynamical equilibrium. Then $N_T$ temperatures $T_i$ from $T_{\min}$ to $T_{\max}$ are chosen (they does not need to be uniformly separated).

A single configuration $\mathcal{C}$ of $L^2$ random variables $\theta$ is initiated. Then, starting at $T_m = T_0 = T_{\max}$, we have

1. **Do** $M_{\mathrm{th}}$ Monte-Carlo steps for the thermalization via the chosen algorithm at temperature $T_m$, starting with the configuration $\mathcal{C}$;

2. With the last configuration $\mathcal{C}_{M_{\mathrm{th}}}$ obtained after from the previous step, **do** $M$ Monte-Carlo steps for the measurement process. At each one of the steps, calculate the observables $O_m$. Store these values in an array of size $M$.

3. Store the final configuration, $\mathcal{C}_m$, as a representative of this temperature.

4. Lower the temperature $T \to T_{m+1}$, and start back on step 1, with the initial configuration $\mathcal{C} = \mathcal{C}_m$;

5. For each temperatures, calculate the thermodynamical averages of the measured quantities, such that $\langle O \rangle = \frac{1}{M} \sum_a O(a)$.

The number of thermalization and measurement steps is left to the individual performing the simulation to choose with respect to its device. The larger $M$ is, the longer simulations will be, but one will obtain smaller error bars. Generally, one abides by $M = M_{\mathrm{th}} \sim 10^5$ updates per individual spin (so $M \sim 10^5 \times L^2$ if one uses the local Metropolis update). A simple Python code for this protocol, using the Metropolis and the Wolff algorithm, can be found on my Github.

## 4.5   Statistical analysis: Jackknife method

Performing thermodynamical *averages* with the Markov-Chain Monte-Carlo process is all well and good, but there is one essential element we have swept under the rug. In fact, after a single MC update, the configuration $\mathcal{C}_a$ obtained is not that much statistically different from the one that preceded it $\mathcal{C}_{a-1}$, and the ones before that. Therefore, when we are performing $\langle O \rangle = \frac{1}{M} \sum_a O(a)$, we are summing over a lot of statistically similar configurations.

If we had uncorrelated measurement, like a purely random set of configurations, then this average is valid and the variance associated with this result is $\sigma_O^2 = (\langle O^2 \rangle - \langle O \rangle^2)/N$. In such a definition, we then can provide an estimate that (assuming a Gaussian distribution) the true value of $\langle O \rangle$ is, for $\sim 68\%$ of the simulations, within one sigma of it: $\in [\langle O \rangle - \sigma_O, \langle O \rangle + \sigma_O]$. However, as we mentioned, we do not have uncorrelated measurements. In fact, our measurements are pretty correlated! This modifies the equation for the variance, such that the *actual* variance for uncorrelated measurements is:

$$\sigma_{O,\mathrm{real}}^2 = \sigma_O^2 \tau_{O,\mathrm{int}} \,, \tag{89}$$

where $\sigma_O = (\langle O^2 \rangle - \langle O \rangle^2)/N$ is measured over the whole set of configuration. It is our naive estimate. $\sigma_{\bar{O},\text{real}}$ is our estimation of the *true* variance, taking into account the correlation of our data. In this expression, the integrated autocorrelation time has been introduced

$$\tau_{O,\text{int}} = 1 + 2 \sum_{k=1}^{N} A(k)(1 - \frac{k}{N}) , \tag{90}$$

$$A(k) = \frac{\langle O_a O_{a+k} \rangle - \langle O_a \rangle \langle O_a \rangle}{\langle O_a^2 \rangle - \langle O_i \rangle \langle O_i \rangle} , \tag{91}$$

where $\langle O_a O_{a+k} \rangle = \frac{1}{M} \sum_a O(a)O(a+k)$ and $\langle O_a \rangle = \frac{1}{M} \sum_a O(a)$. One can measure this function $A(k)$ for a given type of measurement (like the energy) and extract the autocorrelation time. It is essentially a measure of *how many* MC steps one should take to obtain a new, uncorrelated, statistically different configuration.

For large separations $k$, the autocorrelation function decays exponentially $A(k) \to a e^{-k/\tau_{O,\text{exp}}}$. By fitting this ansatz to the measured autocorrelation, one can find $\tau_{O,\text{exp}}$, which provides an easy estimate for the true error bars of the obtained data:

$$\delta_O = \sqrt{\sigma_0^2 (1 + 2\tau_{0,\text{exp}})} . \tag{92}$$

In Fig 6 we show the autocorrelation function $A(k)$, and the extracted $\tau_{0,\text{exp}}$ for the XY model in its high temperature phase, using the Wolff algorithm. It can readily be seen that the autocorrelation time is longer as temperature is reduced. This is the phenomena of critical slowing down [23]. As one near a phase transition, or any critical regime, the correlation length $\xi$ diverges. It means that it will take more and more local updates before a new configuration that has changed roughly $\xi^2$ sites is achieved. Only then can a statistically different configuration and measurement obtained. Critical slowing down leads to the big appeal of global update. They still have critical slowing down, the it is much better, *i.e.* the autocorrelation times are smaller than for local updates.

One can actually bypass this entire autocorrelation issue by calculating the averages and variances differently. Here we will use the jackknife technique, though others, such as the binning and bootstrap method, exist. In the jackknife analysis, one splits the $M$ measurements of $O$ into $N_J$ equal sized Jackknife blocks. In each block, there is $k$ measurements, such that $M = k N_J$. Then, a set of measurement $O_{J,n}$ with $n = 1, \cdots, N_J$ is constructed, which contains all the data except the $n^{\text{th}}$ Jackknife block. Then

$$O_{J,n} = \frac{M\bar{O} - kO_n}{M - k} \qquad O_n = \frac{1}{k} \sum_{i=1}^{k} O_{(n-1)k+i} ,$$

$$\bar{O} = \frac{1}{M} \sum_{i=1}^{M} O_i \qquad \bar{O}_J = \frac{1}{N_J} \sum_{n=1}^{B_j} O_{J,n} . \tag{93}$$

One then has the following estimate for the error:

$$\sigma_{\bar{O},\text{real}}^2 = \frac{N_J - 1}{N_J} \sum_{n=1}^{N_J} (O_{J,n} - \bar{O}_J)^2 . \tag{94}$$

This takes into account the effect of correlated measurements and renders a more accurate estimation of the variance and therefore of the error bars for our measurements.

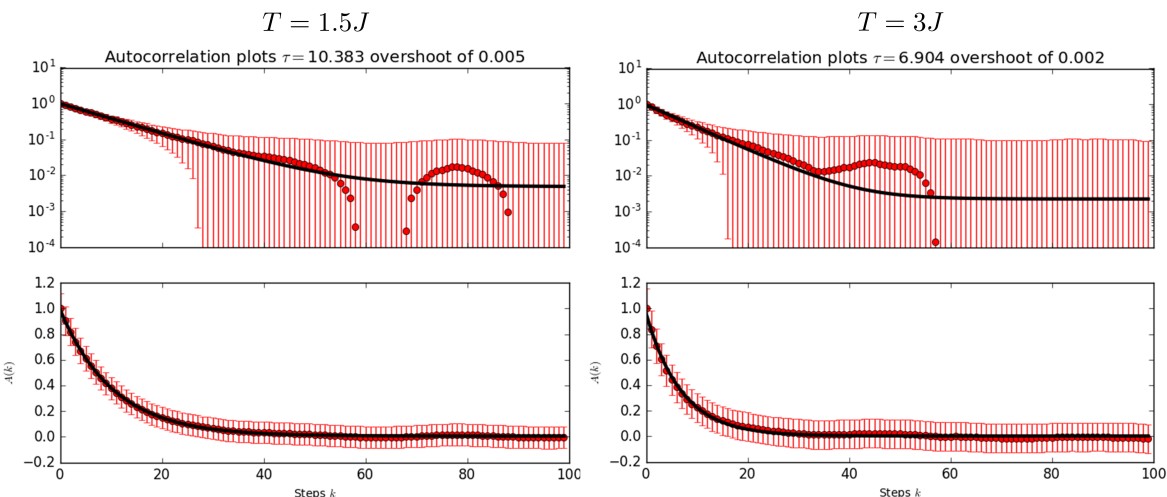

Figure 6: The autocorrelation function $A(k)$ for the energy for 100 MC steps, in a log plot (top) and linear plot (bottom). On the left is $T = 1.5J$ and on the right, $T = 3.0J$. Simulations performed for a $L = 6$ lattice using the Wolff algorithm. A fit (black line) is shown to the form $ae^{-k/\tau} + c$, from which we can estimate the integrated autocorrelation time $\tau_{E,\exp}$. )

## 4.6 Numerical evidence for the KT transition

Before showing some Monte-Carlo results fr the XY model, two important notions need to be set it. Firstly, in typical second-order phase transitions, the observation of the specific heat or the susceptibility would suffice to pinpoint the critical temperature. Indeed, as one increases the size $N$ of the simulated systems, these two quantities will have a power-law divergence at the critical temperature. As we explained in the beginning of these notes, this is not the case for the XY model. There will not be any singular behavior in $C_V$ at $T_{KT}$, and instead one observes a broad bump around $T_{cv,max} = 1.1T_{KT}$. This however is not a rigorous statement, simply an observation, and cannot be used to precisely locate a KT transition.

On the other hand, we can use the notion of the Nelson-Kosterlitz jump as a precise criteria for the location of the phase transition. As we have shown above, the spin stiffness $\rho_s$ is the observable related to the effective coupling constant. Therefore, for a given linear system size $L$, one can find $T_{KT}$ through

$$\rho_s(T_{KT}) = \frac{2T_{KT}}{\pi} . \tag{95}$$

The scaling equations for the XY model can also help understand the scaling behavior of the pseudo-critical KT temperature, that is the extracted $T_{KT}(L)$ from the above criteria. Indeed, one finds that is scaled like the inverse of a logarithm [24]

$$T_{KT}(L) = T_{KT}(\infty) + \frac{a}{(\ln L)^2} . \tag{96}$$

This is a very useful tool in extracting the thermodynamic limit of the critical temperature. It needs to be constrasted with second order phase transitions, where scaling of the pseudo-critical temperature gives $T_c(L) = T_{KT}(\infty) + aL^{-1/\nu}$, $\nu$ being one of the critical exponents. This scaling

converges much faster. Indeed, the XY model is plagued by these logarithmic scaling forms, which means that one must reach very large system sizes and push the computational power available to infer accurate results.

Furthermore, the principal tenets of the Kosterlitz-Thouless theory can be checked numerically. For $T = T_{KT} + \epsilon$ (slightly above the critical temperature), the spin stiffness $\rho_s(T, L)$ should tend to 0 as $L \to \infty$, such that $\rho_s(T, L) \sim 1/\ln(L)$. For $T = T_{KT} - \epsilon$, the stiffness becomes finite in the thermodynamic limit $\rho_s(T, L \to \infty) \to \rho_{s,0}(T)$. This is a numerical confirmation of the Nelson-Kosterlitz jump.

## 4.7   Results

We use the Wolff algorithm to simulate the 2D square lattice XY model for system sizes $L \times L$ with $L \in \{10, 20, 30, 40, 60, 80\}$. The range of temperature that was investigated is the relevant temperature range from $T = 0.5J$ to $T = 1.5J$. Errorbars are extracted using the Jackknife method.

The first results are presented in Fig. 7. We find that the specific heat does not seem to diverge at any temperature, and a bump at $T \sim 1.0J$ is seen, heralding the unbinding of the vortices. The magnetization is finite at all temperatures, though it is reduced for increasing $L$ in the low-temperature regime, indicating QLRO. This is associated with a pronounced magnetic susceptibility at $T = T_{KT}$. For $T < T_{KT}$ the susceptibility goes down, but it increases for larger and larger system sizes - indeed, this low temperature phase is critical with infinte correlation length and therefore susceptibility.

In Fig. 8, we first see that the vortex density $\omega_v$ is near zero at low temperature, and for $T > 0.8$ is starts to climb. Finally, the spin stiffness is clearly finite at low temperatures, and tracks well with the dashed line corresponding to the effect of spin waves that reduce the spin siffness. As $T$ increases, $\rho_s$ deviates from the dashed line - this is interpreted as the onset of the presence of vortices in the system, and tracks with the increase of $\omega_v$. These renormalize the effective coupling $\rho$ that now falls below the dashed line.

The stiffness crosses the line $2T/\pi$ at a finite temperature, such that, through the Nelson-Kosterlitz criteria, we can extract $\rho(T_{KT}) = 2T_{KT}/\pi$. For temperatures above this, the stiffness is strongly reduced, and in the thermodynamic limit will go to 0. We can extract a pseudocritical critical temperature $T_{KT}(L)$, and using the scaling relation, fit it to $a/(\ln L)^2$. Finding then the y-intercept leads to our estimate of the critical temperature in the infinite system size to be $T_{KT}(\infty) \sim 0.89$, very close to the result quoted in the literature of $T_{KT} = 0.89213(10)$. In order to get a tighter estimate of the critical temperature, one would need to run this for larger and larger system sizes. System sizes of up to $L \sim 10^3$ has been reached by some groups, which lead to the result quoted above [25]. This requires agressive optimization of the code implementation.

# 5   Universality of the mechanism

In this section, we explore different physical settings where QLRO is attained through a Kosterlitz-Thouless phase transition. These notes has so far focused on the classical magnet setting of planar spins on a square lattice. There are many more physical settings that exhibit the same universal features. We provide insight into the study of superfluidity, where BKT physics is at play notably in the explanation of the sharp drop of the superfluid stiffness in He-4 torsion experiments. We also provide an introduction to the theory of melting of two-dimensional crystals through the

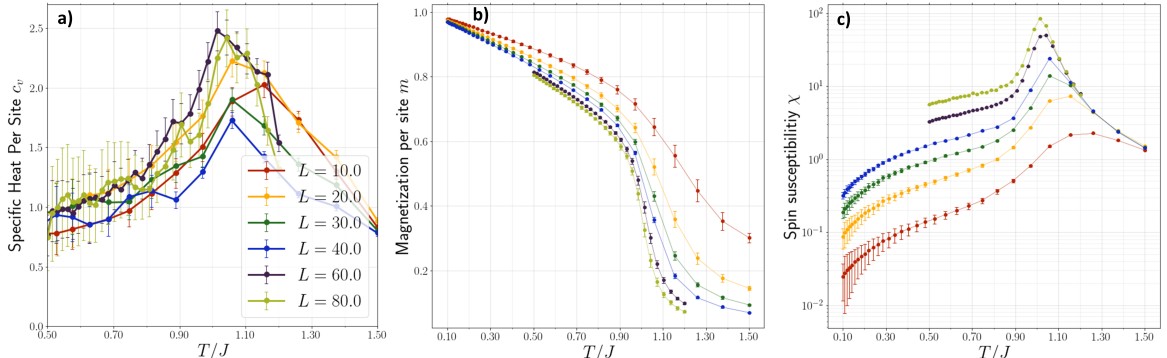

Figure 7: Thermodynamical results for the 2D classical XY model, at different linear system sizes $L$, using the Wolff algorithm: (a) the specific heat per site, (b) the magnetization and (c) the magnetic susceptibility.

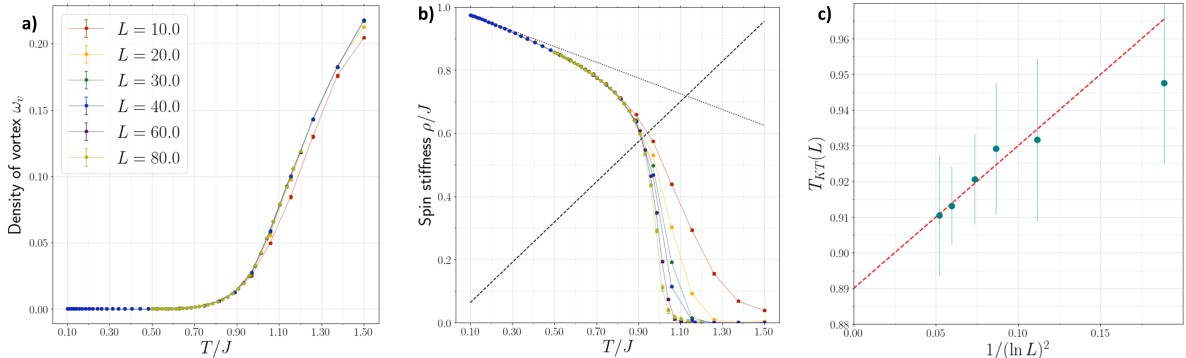

Figure 8: Thermodynamical results for the 2D classical XY model, at different linear system sizes $L$, using the Wolff algorithm: (a) the average density of vortices, (b) the spin stiffness and (c) the extracted pseudo-critical Kosterlitz-Thouless transition temperatures as a function of system size. Red line is a fit for large system sizes that extrapolates to $T_{KT}(L = \infty) \sim 0.89$.

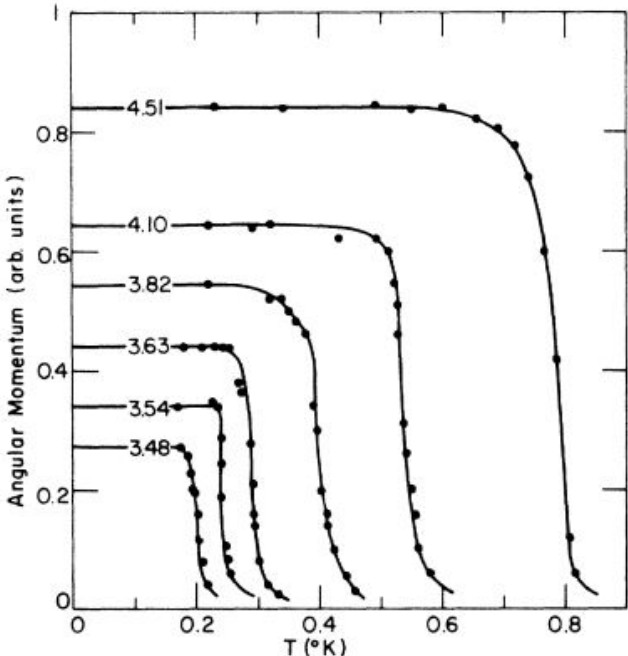

Figure 9: Persistent current angular momentum (which is related to the superfluid density) as a function of temperature for different thin film coverage, i.e. thickness. A discontinuous jump is clearly visible. Taken from Ref. [31].

unbinding of different type of crystal defects. An effective $O(2)$ order parameter can be deduced and these defects interact as in a 2D Coulomb gas, therefore the KT theory can be applied to this setting. These settings were pointed out in the first paper by Kosterlitz and Thouless as potential realizations of the described physics [4, 5].

In both of these cases, one observes experimentally some type of order at low temperature. 2D crystals exhibit what seems to be some translational order, even though the mean square deviation of $\langle u^2(R) \rangle$ of particles from their equilibrium position $R$ can be shown to diverge as $\ln L$ as the system size increases $L$ [26, 27]. In two-dimensisonal superfluids, Hohenberg showed that there cannot be any try Bose-Einstein condensation [28]. These are all restatements of the Mermin-Wagner theorem [1, 29, 30], which we showed above. Kosterlitz-Thouless theory provides an understanding of a true low-temperature phase without long-range order, but one that is distinct from the disordered phase. Therefore, the observation of "ordered" phases in these examples is a testament to the slow algebraic decay of correlations in a quasi-range-ordered system.

## 5.1   Superfluid He-4

Experiments of films of Helium-4 had revealed a discontinuous transition between the normal liquid state and the superfluid state, such that the superfluid density was 0 for $T_c^+$ and if was finite for $T_c^-$. This is shown in Fig. 9. This is undoubtedly not in any second-order phase transition's universality class, which show continuous evolution of order parameters. The absence of a sharp specific heat peak [32] rules out a first-order transition. Therefore a puzzle is left, which the KT theory successfully solves.

He-4 atoms are bosons due to them being composites of an even number of fermions. The

complex order parameter for superfluid Helium-4 is the macroscopic condensate wavefunction $\Psi(r) = \sqrt{\rho(r)}e^{i\theta(r)}$, with $\rho(r)$ being the superfluid density and $\theta(r)$ being the phase. Both can vary in space. The phase $\theta$ is related to the superfluid velocity:

$$\mathbf{v}_s(r) = \frac{\hbar}{m}\nabla\theta(r)\,, \tag{97}$$

with $m$ being the mass of the boson causing the superfluidity. This can be for $He^4$, or can be more general for other bosonic contexts such as a Cooper pair, with $m = 2m_e$. Note that we are talking here about 2D superfluids, which means in reality that the 3rd dimension, *i.e.* the thickness of the layer, is small enough so that there is effectively no superfluid flow in that direction. In that sense, the superfluid density is then an average of the actual density over the thickness of the layer: $\rho(r) = \langle\rho(\mathbf{r})\rangle_z$. In experiments probing thin films of Helium-4, the coverage (density per area) is typically such that at most 2 atomic layers of superfluid are present. The average superfluid density is written as $\rho^0 = \langle\rho(\mathbf{r})\rangle_{xy}$. Without loss of generality, we will take a film in the $x - y$ direction, meaning its normal is $\hat{z}$. The Hamiltonian for such a 2D superluid is identical to the 2D XY model (see Eq. 25):

$$H_S = \frac{\rho_s^0\hbar^2}{2m^{*2}}\int d^2r|\nabla\theta(r)|^2 = \frac{\rho_s^0}{2}\int d^2r\mathbf{v}_s(\mathbf{r})^2\,, \tag{98}$$

where $\mathbf{v}_s(\mathbf{r}) = \frac{\hbar}{m^*}\nabla\theta$ is the superfluid velocity with respect to the lab (or more generally, to the substrate containing the superfluid). Note that this Hamiltonian can alternatively be derived from a Landau-Ginzburg [33] free energy, which described the motion of the condensae with respect to the fluid itself [6]:

$$F[\Psi(r)] = \frac{|\nabla\Psi(r)|^2}{2} + \frac{r(T)}{2}|\Psi(r)|^2 + \frac{u}{4}|\Psi(r)|^4 + \cdots\,. \tag{99}$$

At low-temperatures below the mean-field transition temperature, $r(T) << 0$ and $u >> 0$, therefore amplitude fluctuations of $\Psi(r)$ are very small and can be ignored. We can set $\Psi(r) = \sqrt{-r(T)/u}$. The remaining degrees of freedom are the fluctuations in the phase $\theta(r)$, and we recover the above-mentioned Hamiltonian. Applying the previously derived results from the 2D Coulomb gas duality to 2D superfluids, we get the following expression for the superfluid density

$$\frac{\rho_s^0(T_c)}{T_c} = \begin{cases} \frac{2k_B m^2}{\pi\hbar^2} \text{ for } T \to T_c^- \\ 0 \text{ for } T \to T_c^+ \end{cases}\,. \tag{100}$$

Therefore, the Nelson-Kosterlitz jump is a natural explanation of the discontinuous behavior observed in 2D superfluids, as shown in Fig. 9. Remarkably, one can estimate this number and obtain $3.491 \times 10^{-9}$g cm$^{-2}$K$^{-1}$. Comparing different samples with different $T_c$s in Fig. 10, one sees that they all fall on the same line and the KT theory's prediction of the value of the universal jump is correct. Furthermore, as we mentioned in these notes, the unbinding of vortices produces no singular specific heat peak, and only a broad hump at a temperature slightly above the critical temperature. This lack of singular heat capacity while a sharp discontinuity in the apparent order parameter is seen in experiments [32].

In the experiments presented in the mentioned figures, one prepares a thin sample of Helium-4 coating the surface of a mylar film set in an torsion oscillator [34, 35]. This container is then being externally driven by a torque $\tau(\omega)$ at a certain frequency $\omega$, and the response of the superfluid is

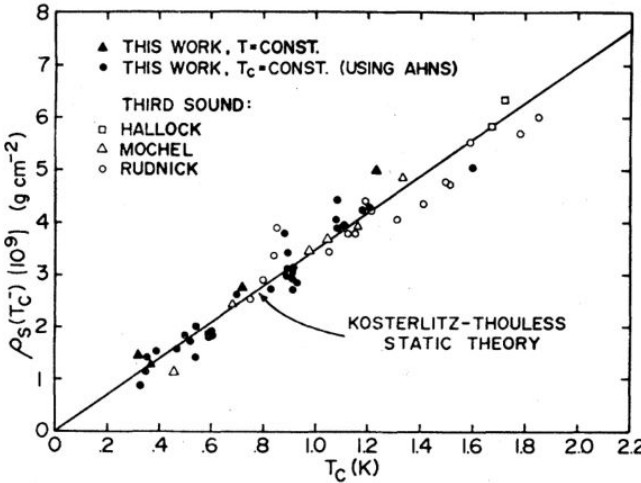

Figure 10: Comparison of different data sets for the value of the superfluid density below $T_c$ as a function of the critical temperature $T_c$. The $T_c$ varies as a function of thin film coverage, as it is seen in Fig. 9.The slope is the static Kosterlitz-Thouless prediction of Eq. 100. Taken from Ref. [35].

measured. A certain angular momentum is imparted to the Helium-4, and when a certain fraction of the sample becomes superfluid, that fraction ceases to participate in the angular momentum. Only the normal fraction imparts angular momentum. Therefore, the anglar moment of inertia is $I(T, \omega) = R^2(M - A\rho_s^0(T, \omega))$, where $R$ is the radius of the container, $A$ its 2D area and $M$ the mass of the container. Since $M \gg A\rho_s$, the shift in the observed period $\Delta P/P_0$ is related to the superfluid density $\rho_s^0(T)$.

One caveat there is that these experiments are always performed at finite frequency, and that the theory we presented here is the static KT theory. In reality, experiments obtain $\rho_s^0(\omega, T)$, and one has to extrapolate the results to $\omega \to 0$ to extract the information of the plots presented above. One can relate this dynamical superfluid density to the dielectric constant of the Coulomb gas of vortices $\epsilon(T, \omega)$, such that $\rho_s^0(\omega, T) = \rho_s^0(T)/\epsilon(T, \omega)$. The detailed inclusion of dynamics within the KT theory, which was due to Ambegaokar et al in 1980 [36], resulted in a series of predictions for the frequency forms of the real and imaginary parts of $\epsilon(T, \omega)$. This is perhaps the strongest test of KT theory, and the agreement is remarkable. These effects are beyond the scope of this text. For further insight into the dynamical KT theory, we suggest Ref. [9] and Tony Leggett's lecture notes to the curious reader.

## 5.2 Two-dimensional melting and KTHNY theory

A solid is described by its rigidity, i.e. an applied stress does not change its shape, and by its crystalline structure, which can be observed by X-ray crystallography. On the other hand, a liquid will flow upon applied stress and is not rigid, as well as being isotropic. This is the typical paradigm of phases of matter in three dimensions, although some examples, like glass, which is clearly rigid yet isotropic, challenge this. In three dimensions, the transition between those two phases is very often a first-order discontinuous transition (like in the water-ice transition) unless fine tuned to a critical point.

On the other hand, in two-dimensions, those two definitions still hold, but one must speak of

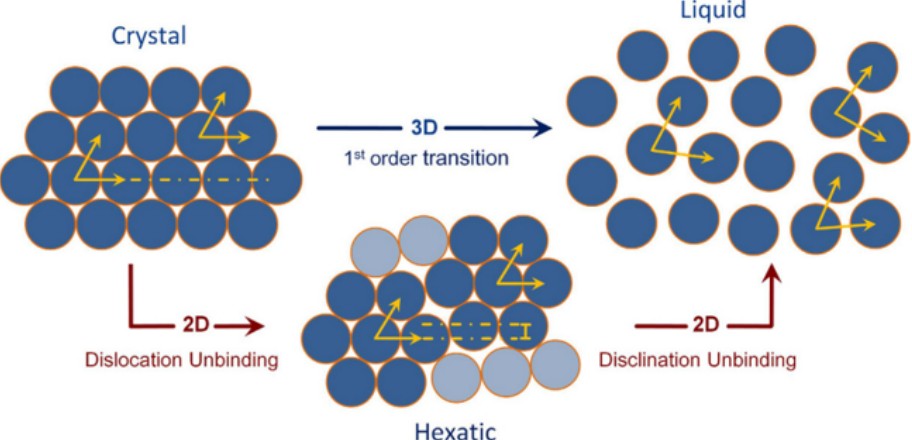

Figure 11: The two melting scenarios for 3D and 2D lattices. The KTHNY scenario is shown as the succession of two KT transitions resulting in an intermediate hexatic phase. Taken from Ref. [40].

quasi-long-range positional order at low-temperature, where the constituents arrange themselves in a lattice. This lattice may melt to the liquid phase, and the process by which it does so has been the subject of a flurry of theoretical studies. In their first papers, Kosterlitz and Thouless first discussed the analogy between the 2D XY models, superfluids and the problem of 2D metling [4,5]. Halperin and Nelson [37,38] as well as Young [39] provided further insight, which resulted in the KTHNY theory of melting.

The KTHNY theory predicts that if the transition from the solid to the liquid is continous, then it will be through an intermediate hexatic phase. This phase has bond-orientational quasi-long-range order, where the bond angle order parameter $\psi$ is described as $\psi = \exp(6i\theta)$. $\theta$ represents the angle that a given hexatic environment around a site has with respect to the lab frame. This quantity has algebraic correlations in the hexatic phase $\langle \psi^*(r)\psi(0) \rangle \sim r^{-\eta(T)}$, while positional correlations are short ranged $\langle u(r)u(r') \rangle \sim \exp(-|r - r'|/\xi)$. In the low-temperature crystalline phase, the hexatic bond-angle parameter is long-range ordered, while the positional correlations are algebraic. This sequence of transitions is shown in Fig. 11. The important insight from KTHNY was in the nature of defects in 2D lattices, and mapping there to effective 2D Coulomb gas Hamiltonians, where the analogy to the KT theory could be completed.

Two types of defects can be identified, which are shown in Fig. 12. Firstly, *dislocations* can occur at low temperatures within a perfect crystal. They can be visualized, as in Fig. 13. These correspond to an extra line within the lattice, and can be represented by a Burger's vector **b**. One can write a modified Coulomb gas Hamiltonian for this **b**, which interact logarithmically (with some modification) with itself at different site. The KT flow equations then lead to the Burger's vectors' stiffness to undergo a sudden jump at a temperature $T_{\text{hex}}$.

Interestingly, a dislocation can be thought of as a bound pair of two *disclinations*, as shown in Fig. 12. Disclinations are defects where the coordination of the lattice differs from that of a perfect lattice. For example, consider the hexatic environment of 6 sites surrounding one site. A defect might show as a site with five or seven neighbors. This introduces a new phase angle $\theta$, which is related to the longitudinal part of the Burger's field **b**. One can write a simple Hamiltonian for $\theta$: $H_{\text{hex}} \sim K_A \int d^2x |\nabla\theta(x)|^2$. Defects in the bond-angle field are the disclinations. In the hexatic phase, with free dislocations, disclinations are bound in pairs. When they unbound, the

a)                      b)                      c)

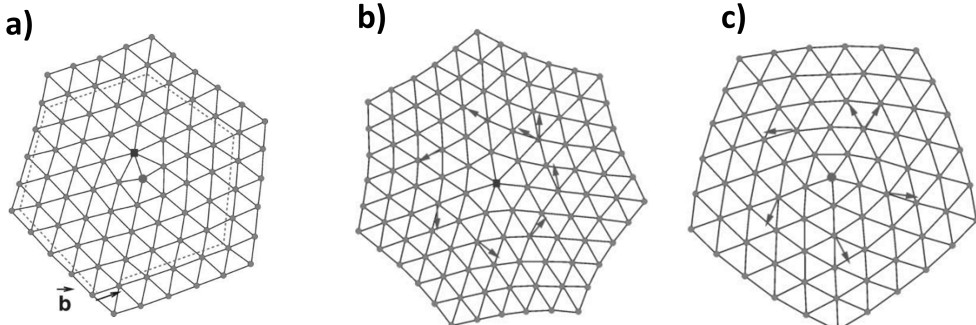

Figure 12: (a) A dislocation in the triangular lattice. As one circles the defect, there is an extra line, characterized by the Burger's vector $b$ that closes the circuit which would occur in a perfect lattice. A dislocation can be thought of as a bound neutral pair of disclinations. (b) and (c) $\pm\pi/3$ disclinations on a triangular lattice. They respectively are points with one seven or five nearest neighbors (as opposed to six) in a perfect lattice. There is an oriental mismatch. Taken from Ref. [41].

stiffness of the bond-angle order parameter $K_A$ discontinuously drops to zero. This is the second Kosterlitz-Thouless phase transition $T_{\text{iso}}$, and at higher temperatures the system is in the isotropic phase.

The diffraction pattern of these three phases is shown in Fig. 14. Electron beams can be sent on films of liquid crystals, which then diffract. Collecting this diffraction pattern helps generate very good scattering plots $S(q)$ in the various phases. One can see that, in the isotropic phase, there is no clear feature. In the hexatic phase, six diffuse spots with algebraic correlations (these are wide Lorentzians) are seen. In the crystal phase, those peaks become Bragg peaks and become very sharp, indicating bond-orientational long-range order.

One thing to note however is that while there exist many experimental settings that vindicate the KTHNY theory, there are also many settings where the transition from the crystalline phase to the liquid occurs through a weak first-order transition. It is now understood that changes in the core energy of vortices, or topological defects, can push the two KT transitions together into a weak 1st order transition [45–47]. Therefore, some variation in the core energy of different types of liquid crystals might put them in different sequences of transitions. Previous work by the author of these notes has been devoted to the study of a puzzling set of experiments in liquid crystals that show a hexatic phase, but also the appearance of a strange third transition that seems to be second-order. The theoretical model explored in Ref. [48] aims at explaining this modified KTHNY scenario, where the presence of *fractional* topological defects is revealed.

We have not mentioned the complex problem of addressing the multi-layer physics of liquid crystals or their smectic behavior, or the interaction with substrates. The physics of liquid crystals, and the adjacent field of soft condensed matter physics, which deals in membranes, coloids, polymers and others settings, is rich in features and novel concepts. For a great introduction to the subject from the statistical mechanics point of view, the excellent book by Chaikin and Lubensky [6] is recommended.

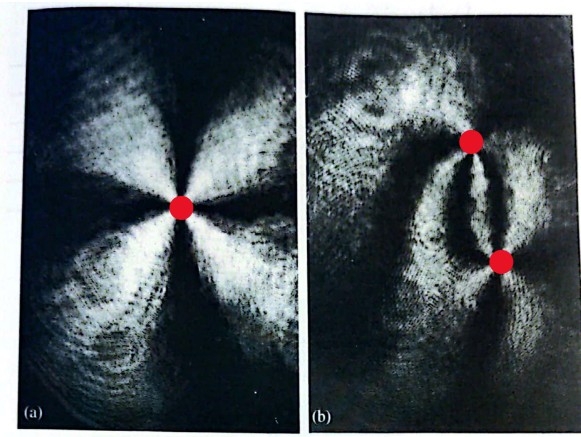

Figure 13: Visualization of a single disclination (left) and a pair of disclinations (right) in a thin film of liquid crystals. A polarized reflection microscope using an argon-ion laser is used as the light source, and as it passes through the sample, different orientational orders lead to different light intensity in the projected image. Taken from Refs. [42].

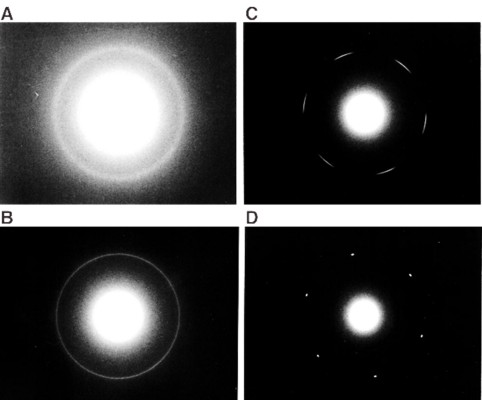

Figure 14: Electron diffraction spectrum $S(q)$ of monolayer thin films of liquid crystal. (a) is a high-temperature isotropic phase, and (b) is still isotropic but closer to the transition to the hexatic phase, where the angular isotropic ring at fixed $q$ can be seen. (c) is the hexatic phase with six diffuse spots describing the quasi-long-range bond-orientational order, (d) the low-temperature solid, with sharp Bragg peaks showing quasi-long-range positional order. Adapted from Ref. [43, 44].

## 5.3   Other settings

In this section, we have not mentioned the application of the KT transition theory to 2D degenerate Bose gases [49, 50], Josephson junction arrays [51], its observation in 2D superconducting thin films [52–54]. We also omitted the mapping of the 2D Coulomb gas partition function to that of the quantum sine-Gordon model [55], with possible applications to quantum circuits [56]. These are all active areas of research, and a testament to the universality of the topological defect unbinding scenario. In the era of quantum computing, quantum information and machine learning, the KT transition even provides a rich testing ground for interesting techniques and concepts, such as quantum annealing of qubit lattices [10] and the machine learning of topological states of matter [57, 58].

## 6   Outlook

In this paper, we covered the basic theory of the classical XY model, *i.e.* its mapping to a 2D Coulomb gas and its renormalization group flow, and recounted the essential elements that lead to the idea of topological defects called vortices. These are bound at low-temperature, and their unbinding leads to the loss of quasi-long-range order. This is a radical departure from the usual second-order phase transition, which is borne out of an explosion of continuous fluctuations. We then showed how the classical XY model can be studied computationally using the Monte-Carlo method, and showed some brief results as to how one extracts the critical temperature from numerical simulations. Finally, we non-exhaustively covered some physical contexts where KT physics applies, due to their planar symmetry and them residing in two-dimensional space.

Interestingly, the KT transition can be thought of as a classical example of **asymptotic freedom** [59, 60]. Indeed, screening of the logarithmic interaction between vortices at high temperatures means that they are asymptotically free, with their fugacity scaling to infinity. On the other hand, at low temperature, the interaction is very weakly screened and vortices rather form neutral pairs.

## Acknowledgements

I am indebted to Piers Coleman, Premi Chandra, Peter P. Orth and Tom Lubensky for their teaching and our collaboration, which set me on the path to dive deep within the XY model literature. I also thank Kun Chen and Ananda Roy for related discussions on the subject.

**Funding information**   This work was done while supported by DOE Basic Energy Sciences grant DE-FG02-99ER45790 and the Fonds de Recherche Québécois en Nature et Technologie.

## A   Derivation of the RG equations from the Coulomb gas picture

In this appendix, we provide an in-depth derivation of the RG equations for the 2D Coulomb gas. Starting from

$$e^{\mathcal{H}_{\text{eff}}(r-r')} = \left\langle e^{-2K\pi\ln(|\boldsymbol{r}-\boldsymbol{r}'|)} \right\rangle_T . \tag{101}$$

We then have:

$$e^{\mathcal{H}_{\text{eff}}(r-r')} = \langle e^{-2K\pi\ln(|\boldsymbol{r}-\boldsymbol{r}'|)}\rangle_T \tag{102}$$

$$= \frac{\sum_{N=0}^{\infty} \frac{y_0^{2N}}{(N!)^2} \int \left(\prod_{i=1}^{2N} \frac{d^2 s_i}{a^2}\right) \exp\left[-2K\pi\ln(|\boldsymbol{r}-\boldsymbol{r}'|) + 2K\pi\sum_{i<j} n_i n_j \ln(|\boldsymbol{s}_i-\boldsymbol{s}_j|)\right]}{\sum_{N=0}^{\infty} \frac{y_0^{2N}}{(N!)^2} \int \left(\prod_{i=1}^{2N} \frac{d^2 s_i}{a^2}\right) \exp\left[2K\pi\sum_{i<j} n_i n_j \ln(|\boldsymbol{s}_i-\boldsymbol{s}_j|)\right]} . \tag{103}$$

As we said, we keep only the $N = 0, 1$ terms, and then have a result up to $O(y_0^4)$. We also, out of simplicity, set $a = 1$. This gives us

$$e^{\mathcal{H}_{\text{eff}}(r-r')}$$

$$= \frac{e^{-2K\pi\ln(r-r')}\left\{1 + y_0^2 \int d^2 s\, d^2 s'\, e^{-2K\pi\ln(s-s')+2K\pi D(r,r',s,s')} + O(y_0^4)\right\}}{1 + y_0^2 \int d^2 s\, d^2 s'\, e^{-2K\pi\ln(s-s')} + O(y_0^4)} , \tag{104}$$

with

$$D(r,r',s,s') = \ln(r-s) - \ln(r-s') - \ln(r'-s) + \ln(r'-s') . \tag{105}$$

Here, we have, encoded into the function $D(r,r',s,s')$, the interaction between our two internal charges and our two external charges. Furthermore, the sign in front of the different logarithmic interactions depends on the product of the charge of the different vortex (ex. $s$ with $r$, same charge, positive sign; $r'$ and $s$, opposite charge, negative sign). We then exploit that $y_0^2 \ll 1$, and can write $(1 + [\cdots]y_0^2 + O(y_0^4))^{-1} = 1 - [\cdots]y_0^2 + O(y_0^4)$. This gives us:

$$e^{\mathcal{H}_{\text{eff}}(r-r')+2K\pi\ln(r-r')} = \left\{1 + y_0^2 \int d^2 s\, d^2 s'\, e^{-2K\pi\ln(s-s')+2K\pi D(r,r',s,s')} + O(y_0^4)\right\}$$

$$\times \left\{1 - y_0^2 \int d^2 s\, d^2 s'\, e^{-2K\pi\ln(s-s')} + O(y_0^4)\right\}$$

$$= 1 + y_0^2 \int d^2 s\, d^2 s'\, e^{-2K\pi\ln(s-s')} \left\{e^{2K\pi D(r,r',s,s')} - 1\right\} + O(y_0^4) . \tag{106}$$

Noting that the prefactor $\exp(-2K\pi\ln(s-s'))$ in the integral will lower dramatically the statistical weight of configurations that have a very large distance $x = s-s'$. In light of that, we would like to work within the center of mass framework, in which we have $x = s-s'$, $X = (s+s')/2$. We then expand our logarithms for small $x$, and dropping the bold expression for vectors (positions $r$ and $s$ remain vectors), we have

$$\ln(r-s) = \ln(r-X) - \frac{x \cdot \nabla_X}{2} \ln(r-X) + \frac{x^2 \nabla_X^2}{4} \ln(r-X) + O(x^3),$$

$$\ln(r-s') = \ln(r-X) + \frac{x \cdot \nabla_X}{2} \ln(r-X) + \frac{x^2 \nabla_X^2}{4} \ln(r-X) - O(x^3),$$

$$\ln(r'-s) = \ln(r'-X) - \frac{x \cdot \nabla_X}{2} \ln(r'-X) + \frac{x^2 \nabla_X^2}{4} \ln(r'-X) - O(x^3), \quad (107)$$

$$\ln(r'-s') = \ln(r'-X) + \frac{x \cdot \nabla_X}{2} \ln(r'-X) + \frac{x^2 \nabla_X^2}{4} \ln(r'-X) - O(x^3).$$

We then obtain a new expression for $D$:

$$D(r, r', s, s') = -x \cdot \nabla_X \ln(r-X) + x \cdot \nabla_X \ln(r'-X) + O(x^3). \quad (108)$$

Therefore, we can also expand the last term in eq. 106 to third order in $x = s-s'$, the separation of the two internal charges, still having $X = (s+s')/2$ the center of mass of the internal vortex-antivortex pair. Using $e^x = 1 + x + x^2/2 + O(x^3)$, we have

$$e^{2K\pi D(r,r',s,s')} - 1 = -2K\pi \, x \cdot \nabla_X[\ln(r-X) - \ln(r'-X)]$$
$$+ 2K^2\pi^2[x \cdot \nabla_X(\ln(r-X) - \ln(r'-X))]^2 + O(x^3). \quad (109)$$

The next steps are far trickier. First, we plug back eq. 109 into eq. 106, and start integrating the $s$ and $s'$ degrees of freedom. However, it is easier to change from $d^2s\,d^2s'$ to a measure in $x$ and $X$. Calculating the Jacobian, one gets that $d^2s\,d^2s' = |J(x,X)|d^2x\,d^2X = d^2x\,d^2X$. Therefore, one writes:

$$e^{\mathcal{H}_{\text{eff}}(r-r')} = e^{-2K\pi \ln(r-r')}\Big[ 1 + y_0^2 \int d^2s\,d^2s' e^{-2K\pi \ln(s-s')}$$

$$\Big\{ -2K\pi \, x \cdot \nabla_X[\ln(r-X) - \ln(r'-X)]$$

$$+ 2K^2\pi^2[x \cdot \nabla_X(\ln(r-X) - \ln(r'-X))]^2 \Big\} + O(y_0^4) \Big] \quad (110)$$

$$= e^{-2K\pi \ln(r-r')}\Big[ 1 + y_0^2 \int d^2x\,d^2X e^{-2K\pi \ln x} \quad (111)$$

$$\Big\{ -2K\pi \, x \cdot \nabla_X[\ln(r-X) - \ln(r'-X)]$$

$$+ 2K^2\pi^2[x \cdot \nabla_X(\ln(r-X) - \ln(r'-X))]^2 \Big\} + O(y_0^4) \Big]. \quad (112)$$

We first see that the integration of the first term, linear in $x$, goes to 0. Explicitly, we have:

$$\int d^2x\,d^2X e^{-2K\pi \ln x} x \cdot \nabla_X[\ln(r-X) - \ln(r'-X)]$$

$$= \Big\{ \int d^2x \, e^{-2K\pi \ln x} x \Big\} \cdot \Big\{ \int d^2X \nabla_X[\ln(r-X) - \ln(r'-X)] \Big\}, \quad (113)$$

and the integration of the gradient term leads to zero. We then have the second integral left. We first perform the angular integral in $x$, $\int d\theta_x$ such that $d^2x = x dx d\theta_x$. For simplicity's sake, we define $C(X) = [\ln(r-X) - \ln(r'-X)]$.

$$2K^2\pi^2 \int d^2x \, d^2X e^{-2K\pi\ln x}[x \cdot \nabla_X C]^2$$

$$= 2K^2\pi^2 \int (x dx)e^{-2K\pi\ln x} \int d^2X \int d\theta_x[x \cdot \nabla_X C]^2 \,, \tag{114}$$

and then, using $\int_0^{2\pi} d\theta_x \sin^2(\theta_x) = \int_0^{2\pi} d\theta_x \cos^2(\theta_x) = \pi$,

$$\int d\theta_x[x \cdot \nabla_X C]^2 = \int d\theta_x[x \cos\theta_x(\nabla_X C)_1 + x \sin\theta_x(\nabla_X C)_2]^2$$

$$= \int d\theta_x x^2 \big[\cos^2\theta_x(\nabla_X C)_1^2 + 2\sin\theta_x \cos\theta_x(\nabla_X C)_1(\nabla_X C)_2$$

$$+ \sin^2\theta_x(\nabla_X C)_1^2\big] \tag{115}$$

$$= x^2[\pi(\nabla_X C)_1^2 + 2(0)(\nabla_X C)_1(\nabla_X C)_2 + \pi(\nabla_X C)_1^2]$$

$$= \pi x^2(\nabla_X C)^2 \,,$$

leading to

$$2K^2\pi^2 \int d^2x \, d^2X e^{-2K\pi\ln x}[x \cdot \nabla_X C]^2 = 2K^2\pi^3 \int (x dx)e^{-2K\pi\ln x} \int d^2X x^2(\nabla_X C)^2 \,. \tag{116}$$

Then, our expression for the effective interaction is

$$e^{\mathcal{H}_{\text{eff}}(r-r')}$$

$$= e^{-2K\pi\ln(r-r')}\left[1 + y_0^2 2K^2\pi^3 \int (x dx)e^{-2K\pi\ln x} \int d^2X x^2(\nabla_X C)^2 + O(y_0^4)\right]. \tag{117}$$

Noting that $\nabla^2 \ln r = 2\pi\delta^2(r)$, we can calculate the $X$ integral in Eq. 117 by parts, which leads to the cancellation of the infinite surface part

$$\int d^2X(\nabla_X[\ln(r-X) - \ln(r'-X)])^2$$

$$= \big\{[\ln(r-X) - \ln(r'-X)] \times \nabla_X[\ln(r-X) - \ln(r'-X)]\big\}\big|_{\mathcal{S}_X \to \infty}$$

$$- \int d^2X[\ln(r-X) - \ln(r'-X)] \times (\nabla_X^2[\ln(r-X) - \ln(r'-X)]) \tag{118}$$

$$= -\int d^2X[\ln(r-X) - \ln(r'-X)] \times (2\pi\delta^2(r-X) - 2\pi\delta^2(r'-X))$$

$$= -2\pi(\ln(r-r) - \ln(r'-r) - \ln(r-r') + \ln(r'-r'))$$

$$= 4\pi(\ln(r-r') - \ln(0)) \,. \tag{119}$$

The $\ln(0)$ term is a short-distance divergence, which we remove by using the natural dutoff of the system: the lattice size $a$. This leads to a change $\ln x \to \ln(x/a)$. This fixes the bounds of the $x$ integral to $\int_1^\infty$ rather than $\int_0^\infty$, and the second term of Eq 119 is 0. We then obtain:

$$e^{\mathcal{H}_{\text{eff}}(r-r')}$$

$$= e^{-2K\pi\ln(r-r')}\left[1 + 8K^2\pi^4 y_0^2 \int_1^\infty dx\, x^3 e^{-2K\pi\ln x}(\ln(r-r')) + O(y_0^4)\right] \tag{120}$$

$$= e^{-2K\pi\ln(r-r')}\left[1 + 8K^2\pi^4 y_0^2 \ln(r-r')\int_1^\infty dx\, x^{3-2K\pi} + O(y_0^4)\right] \tag{121}$$

$$= e^{-2K\pi\ln(r-r')}e^{8K^2\pi^4 y_0^2\ln(r-r')\int_1^\infty dx\, x^{3-2K\pi}+O(y_0^4)}, \tag{122}$$

where the last step is made by recognizing that the two terms look like the Taylor serie of $e^x$. This is the result quoted in Eq.42 in the main text, which leads to

$$K_{\text{eff}} = K - 4\pi^3 K^2 y_0^2 \int_1^\infty dx\, x^{3-2\pi K} + O(y_0^4). \tag{123}$$

We can then split this integral in two, up to a finite distance $b$.

$$K_{\text{eff}} = K - 4\pi^3 K^2 y_0^2 \left\{\int_1^b dx\, x^{3-2\pi K} + \int_b^\infty dx\, x^{3-2\pi K}\right\} + O(y_0^4). \tag{124}$$

Using the fact that $y_0$ is very small (it is, after all, our expansion parameter), we invert our equation 124. We then have

$$K_{\text{eff}}^{-1} = K^{-1}(1 - Ky_0^2 \int [\cdots] + O(y_0^4))^{-1}$$

$$= K^{-1}(1 + Ky_0^2 \int [\cdots] + O(y_0^4)) \tag{125}$$

$$= K^{-1} + y_0^2 \int [\cdots] + O(y_0^4).,$$

$$\Rightarrow K_{\text{eff}}^{-1} = K^{-1} + 4\pi^3 K^2 y_0^2 \int_1^\infty dx\, x^{3-2\pi K} + O(y_0^4). \tag{126}$$

The use of eq. 126 is that we can then rewrite eq. 124 as

$$K_{eff}^{-1} = \widetilde{K}^{-1} + 4\pi^3 y_0^2 \int_b^\infty dx\, x^{3-2\pi\widetilde{K}} + O(y_0^4), \tag{127}$$

$$\Rightarrow \widetilde{K}^{-1} = K^{-1} + 4\pi^3 y_0^2 \int_1^b dx\, x^{3-2\pi K} + O(y_0^4). \tag{128}$$

Eq. 128 now gives us an effective $\tilde{K}$ for the scale $b$. We now seek to rescale Eq 127 into it's original shape, as in Eq.126 (with $\widetilde{K}$ everywhere). This can be achieved by the simple rescaling

$x \to x/b$. This gives us the same equation as before, but now with rescaled $\widetilde{K}^{-1}$ and $\tilde{y}_0$. The effect of the rescaling on those gives us:

$$\widetilde{K}^{-1} = K^{-1} + 4\pi^3 \tilde{y}_0^2 \int_{1/b}^{1} dx \, x^{3-2\pi K} \,, \tag{129}$$

$$\tilde{y}_0 = b^{2-\pi K} y_0 \,. \tag{130}$$

Choosing an infinitesimal renormalization parameter $b = e^l \approx 1+l$, one can obtain differential equations for the running coupling parameters $K^{-1}$ and $y_0$ in the limit $l \to 0$. If we identify $\widetilde{K}^{-1} = K^{-1}(l)$, $K^{-1} = K^{-1}(0)$ and $\tilde{y}_0 = y_0(l)$, $y_0 = y_0(0)$, we have:

$$\frac{dK^{-1}}{dl} = \lim_{l \to 0} \left[ \frac{\widetilde{K}^{-1} - K^{-1}}{l} \right] \qquad\qquad \frac{dy_0}{dl} = \lim_{l \to 0} \left[ \frac{\tilde{y}_0 - y_0}{l} \right] . \tag{131}$$

For $\tilde{y}_0 = b^{2-\pi K} y_0$, an infinitesimal $b$ leads to

$$\tilde{y}_0 = (1+l)^{2-\pi K} y_0 = y_0 (1 + l(2 - \pi K)) \,,$$
$$\Rightarrow \quad \frac{\tilde{y}_0 - y_0}{l} = (2 - \pi K) y_0 \,. \tag{132}$$

Also, we explicitly perform the integral in Eq. 129, and use our infinitesimal expansion for $b$. This leads to:

$$\begin{aligned}
\widetilde{K}^{-1} &= K^{-1} + 4\pi^3 \tilde{y}_0^2 \int_{1/b}^{1} dx \, x^{3-2\pi K} \\
&= K^{-1} + 4\pi^3 \tilde{y}_0^2 \left[ \frac{x^{4-2\pi K}}{4 - 2\pi K} \Big|_{1/b}^{1} \right] \\
&= K^{-1} + 4\pi^3 \tilde{y}_0^2 \left[ \frac{1^{4-2\pi K} - (1/b)^{4-2\pi K}}{4 - 2\pi K} \right] \\
&= K^{-1} + 4\pi^3 \tilde{y}_0^2 \left[ \frac{1 - (1-l)^{4-2\pi K}}{4 - 2\pi K} \right] \\
&= K^{-1} + 4\pi^3 \tilde{y}_0^2 \left[ \frac{1 - (1 - l(4 - 2\pi K))}{4 - 2\pi K} \right] \\
&= K^{-1} + 4l\pi^3 \tilde{y}_0^2 \,, \\
\Rightarrow \quad \frac{\widetilde{K}^{-1} - K^{-1}}{l} &= 4\pi^3 \tilde{y}_0^2 \,.
\end{aligned} \tag{133}$$

Using equations 132 and 133 within 131, together with the fact that $\lim_{l \to 0} \tilde{y}_0 = y_0$, we get our final flow equations:

$$\frac{dK^{-1}}{dl} = 4\pi^3 y_0^2 + O(y_0^4) \,,$$
$$\frac{dy_0}{dl} = (2 - \pi K) y_0 + O(y_0^3) \,. \tag{134}$$

This is indeed the result quoted in equation 44 in the main text.

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
