# Peer review of "The Kosterlitz-Thouless phase transition: an introduction for the intrepid student"

_SciPost Physics Lecture Notes_

## Round 1 · Referee Report · Dirk Schuricht · 2022-8-17

Report
The author provides a set of lecture notes on the KT transition. This is done with the classic example of the 2D XY model. The underlying physics (vortices), RG technique and Monte-Carlo simulations are well presented. Furthermore, in the last section details on two specific realisations are presented. Overall I find the notes to be very useful for junior researchers as an overview for the topic. If more technical details are required, the reader would have to resort to other references. Given this, I leave it open whether the manuscript is is regarded as suitable for SciPost lecture notes or not.
Th presentation is fine provided the following points are addressed:
-Ref. [2] is mislieading, as it does not contain the 2D transition temperature. Maybe a textbook like Mussardo, Statistical field theory, would be appropriate.
-A bracket is missing in (2).
-I suppose the i without the dot denotes the imaginary unit?
-Please explain why the integral in (9) is one-dimensional. Furthermore, it seems the lower limit is mistyped.
-In the argument leading to (18), where does it go in that the temperature is sufficiently low? Maybe type out \eta(T) for clarity.
-Is the \eta in (19) the same as in (18)?
-Typo in (20).
-In the text above (22), specify what is r.
-Below (24), why is n set to one? \Delta G should probably read \Delta F.
-In the first paragraph of Sec. 3, there appears an slog, which probably should read log.
-The last term in (32) should contain a \nabla^2.
-Second paragraph below (37): Please add brackets around -\beta E_c.
-Above (44), the link is missing.
-The first paragraph of Sec. 3.3 contains the claim that a “rigorous proof” has been given. In my opinion this is not the case, as several approximations have been done in the derivation.
-Please add some explanation why the renormalisation of the fugacity to infinity is unphysical.
-Around (47), explain what the parameter b is.
-At the beginning of page 15, the trace is taken over all states of the system. It is unclear what the given temperature here means.
-In (53) and above I assume that C_1 should read C_{a+1}, ie, the subscript is mistyped.
-After (66), I suppose M goes to zero as a function of T, right?
-Below (67) the subscripts of the field \theta are mistyped.
-Why is there a minus sign in (69), in contrast to (2). This may feed through (70) and (71).
-Paragraph below (75): It seems the last equation contains a typo.
-In the text below (96), maybe T_{KT}(\infty) should read T_c(\infty).
-In Fig. 9, please refer to the main text in the caption. Also, what is the unit for the thickness?
-At the end of the penultimate paragraph of SEc. 5.1, why is M required to be large compared to A\rho_s. It seems that M=const is the case.
-In Sec. 5.3, the appearance of the KT transition in quantum phase transitions like in the XXZ Heisenberg chain may be mentioned. See, eg, Giamarchi, Quantum physics in one dimension.
-In general, there are some typos in the texts and the use of language could be improved (eg, “vanish” instead of “get null”, avoid “kill”).
-Some figures (4, 7, 8) are hard to see and read, ie, the quality should be improved.

---

## Round 1 · Referee Report · Bernard Nienhuis · 2022-9-22

Strengths
1. It brings together a number of important and well-known results on the 2D XY model.
Weaknesses
1. A good introduction of the subject appropriate for the intended readership is lacking
2. In the derivations the choice between what is made explicit and what is left to the reader is less than optimal.
3. Many key concepts, such as universality, long range order, quasi-long range order, are not explained.
Report
The stated goal of the paper is a worthwhile undertaking. Since the stated goal is pedagogical, and and SciPost considers the text for publication, I take it that approval by SciPost means it considers the text good teaching.
Reading through the notes, I gather that the text is meant for students with a good basic knowledge of phase transitions other than the KT transition. In my view the primary literature serves this group relatively well.
A significant improvement can be expected if the text is used as the basis of a course, and the the text is improved with the criticism of the students. I do not think SciPost should be the vehicle to authorize the text as it is now.
The total setup as the sequence of subjects treated seems natural. But the text would gain significant audience if an explanatory introduction would be added. It should be readable for the uninitiated, and explain why the problem is interesting.
The current, brief introductory paragraphs serve well to stop the readers who know the subject well enough, but does not help the readers for whom the text is intended.
The notion of universality class is not explained, but presumed known. It is further confused by the statement on page 3 that the XY transition does not fall in any 2nd or 1st order universality class, which contradicts the claim that it is one of the best examples of universality (above eq. (1)).
On pave 3 the text speake of a low-temperature expansion. Not only is this concept left without explanation, also there is no expansion in sight.
The calculation of the average magnetization is less than precise. Not very inviting for a pedagogical text. What, for instance is the meaning of cos(0) in eq.(5)? The unusual habit to write the square root of -1 as an i without a dot deserves at least to be mentioned.
Eq. (8) apparently requires 'some algebra', not given. I do not find it obvious what algebra is meant. The end result is an expression that only vanishes in the thermodynamic limit. A finite system would thus have a non-zero mean magnetization.
The derivation of the correlation function is less problematic, but here the notation is less than clear. In eq. (11) it is not clear where the argument of $\prod_k$ ends, especially in the last two lines, where it seems to be interrupted by the exponent. Perhaps the author was unable to choose between $\prod_k \exp$ and $\exp \sum_k$, and kept both.
From the context it appears that the sum on k with a prime is the summation over half the Briouillin zone, but that is nowhere stated explicitly.
The derivation (13)-(17) appears unhelpful to me. It would be more helpful to explain in elementary terms where the logarithmic dependence on r comes from, rather than take the route via an explicit Bessel function, which is not used in the end. Also, it is said the integral can be performed in polar coordinates, but the integration is left till the end.
A few typos I came across:
page 2, two lines above eq (1): example -> examples
page 2 line 9: is it really the intention to use 'transition' as a verb?
page 6 following (22) that is a -> that in a
Strengths
The pedagogical and detailed presentation of Kosterlitz RG analysis is valuable
Weaknesses
The quality of the MC results could (should) be improved and the section on applications of BKT transition in Physics is very "classical".
Report
The idea to write a review on the Berezinskii-Kosterlitz-Thouless transition for SciPost Lecture Notes is probably good, but in my opinion, the present paper does not meet the criteria of the journal. This is not a bad idea to present in detail the calculations behind the problem, and the intention of the author is to address students, like the title of the article says, but I think that the paper is still far from being publishable.
A first critique concerns the form of the paper. In my opinion, it is not written in a "professional manner". The style is very relaxed, which is not a default in itself, but it is too much for my taste. Usually, scientific papers are written in an impersonal form. For example the massive use of the possessive "our" makes the text look like a kind of a talk (our equation, our Hamiltonian, our partition function, our integral, our analysis, our decomposition etc). I would say that nothing is ours in science! Even when one speaks about Schr\"odinger equation, it is not called this way because it belongs to Schr\"odinger, but rather because it was discovered by him.
Then, if the literature which is cited is relevant, I think that other prominent contributions would deserve also to be cited. Being a review, and being addressed to students, care should be made at properly citing books (e.g. Kardar's second volume, Itzykson and Drouffe's first volume, Herbut's "modern approach to critical phenomena", even Kleinert's "Gauge fields in condensed matter", ...) and other important reviews (e.g. those of Nelson collected in his "Defects and geometry in condensed matter physics").
Now, my main concern is twofold and is about the two main purposes of this work. One is about the numerical results presented in the paper and the second is about the applications of BKT transition in Physics.
Concerning the numerical results, although the discussion on MC algorithms (sections 4.1 and 4.2) is very clear, the author has chosen to present his own results, but in my opinion they are not "the state of the art" as numericists say. The use of Wolff algorithm is not bad of course, but as the author notices, worms algorithms would be more efficient, and he doesn't use them. Even tensor networks have been recently used for the XY model (https://arxiv.org/abs/1907.04576). Another concern is the quality of the results presented. The sizes reached are not spectacular (not to say that they are small), the material presentation of the plots is not as one could expect (largest sizes have less temperature, which truncates the curves. Of course this is just visually unsatisfactory, but only a small effort would have been needed to really improve this. On the other hand the curves for the specific heat are far from "stabilized" with still strong fluctuations). Another concern with the numerical results is the discussion of the thermalization phase of the runs. An empirical criterion is given ($10^5$ updates per spin) and this is not compared to the autocorrelation time which is yet discussed then. In particular, the same criterion cannot be applied in the vicinity of a "critical slowing down regime" and at high temperatures for example. It gives me the impression that the simulations were performed just as illustrations of the main purpose of the paper, but I think that a review, and this journal, both deserve really higher quality results.
Now concerning the applications of BKT scenario in various areas of Physics, I have to say I am disappointed. Only two applications are discussed, which were already discussed in the original papers by Kosterlitz and Thouless. I would have appreciated for example a discussion of potential applications in cold atoms experiments, BEC-BKT transitions, etc. The 2016 review of Kosterlitz goes far beyond the present paper.
Finally, there are many typos that a careful reading would have eliminated.
E.g.
Eq (19), in th exponential regime of the correlation function, the algebraic prefactor is generically some power, $1/r^p$, with $p$ not necessarily $d-2+\eta$.
$==$ in Eq 20.
$\Delta F$ in eq 24 and $\Delta G$ in the paragraph below.
$q_1q_2s$ p8.
infinity,and p9.
$a$ is the lattice spacing is said p9, but was said before already.
the use of a star on p11 to denote ordinary numbers (integers in fact) product is an unfortunate choice.
is showed
the flow tell us
a criteria
spce
ant
glogal,
and many others, even in formulas ($\theta_ij$ for $\theta_{ij}$ for example).
In conclusion, I am not sure that this paper deserves to be published in SciPost Lecture Notes. At least not in the present form and an important work is required two improve it. In the present form, it does not really bring more than textbooks, or the review paper by Kosterlitz in 2016 in particular. By the way, this short discussion in Ginzburg-Landau theory in the section on He-4 is borrowed to the Kosterlitz review, but with additional errors (there, $\Psi$ is the complex wave function, not the amplitude, but of course, this is the amplitude which equals $\sqrt{-r(T)/u}$ within Landau theory. Things are a bit mixed up).
Requested changes
Improve the MC results, maybe rewrite in a more "professional" style in the spirit discussed above, extend the last section to more modern applications?

---

## Editorial Decision

unknown